# Assessing the Statistical Uniqueness of the Younger Dryas: A Robust Multivariate Analysis

Henry Nye[1] and Alan Condron[2]

[1]Department of Mathematics and Statistics, Haverford College, PA 19041
[2]Geology and Geophysics, Woods Hole Oceanographic Institution, MA 02543

**Correspondence:** Henry Nye (hnye@haverford.edu), Alan Condron (acondron@whoi.edu)

**Abstract.** During the last glacial period (c. 120-11 kyr BP), dramatic temperature swings, known as Dansgaard-Oeschger (D-O) events, are clearly manifest in high resolution oxygen isotope records from the Greenland ice sheet. Although variability in the Atlantic Meridional Overturning Circulation (AMOC) is often invoked, a unified explanation for what caused these 'sawtooth shaped' climate patterns has yet to be accepted. Of particular interest is the most recent D-O shaped climate pattern that occurred from ∼14,600 to 11,500 years ago - the Bølling/Allerød (BA) warm interstadial and the subsequent Younger Dryas (YD) cold stadial. Unlike earlier D-O stadials, the YD is frequently considered a unique event, potentially resulting from a rerouting and/or flood of glacial meltwater into the North Atlantic or a meteorite impact. Yet, these mechanisms are less frequently considered as the cause of the earlier stadials. Using a robust multivariate outlier detection scheme - a novel approach for traditional paleoclimate research - we show that the pattern of climate change during the BA/YD is not statistically different from the other D-O events in the Greenland record, and that it should not necessarily be considered unique when investigating the drivers of abrupt climate change. Our results thus confirm the ambiguity of the BA/YD's trigger and present a novel statistical framework for paleoclimatic data analysis.

## 1 Introduction

First noted in 1985 by Willi Dansgaard as "violent oscillations" in Greenland's DYE-3 and Camp Century oxygen isotope ($\delta^{18}$O) records, Dansgaard-Oeschger (D-O) events are now well known examples of abrupt climate change during the last glacial period (c. 120-11 ky BP) (see Figure 1) (Dansgaard, 1985). These events are characterized by abrupt warmings of ∼ 8-16° C, a subsequent centennial-to-millennial length period of relative warmth (i.e. an interstadial), followed by a gradual, and sometimes abrupt, shift to cooler (stadial) conditions (Li and Born, 2019). Since their discovery three and a half decades ago, countless mechanisms have been proposed to explain D-O cycles (see Li and Born (2019) for a review). Although most hypotheses invoke changes in ocean heat transport as a consequence of variations in the strength of the Atlantic Meridional Overturning Circulation (AMOC) (Broecker et al., 1985), it remains poorly understood as to what would have caused the

overturning cell to undergo such large and rapid changes (Lohmann and Ditlevsen, 2019). While variations in atmospheric circulation, sea ice extent, and ice shelf formation/collapse have all been hypothesized as triggers (Li and Born, 2019), a unifying theory has yet to emerge (Lohmann and Ditlevsen, 2018). Given that D-O events provide compelling evidence that the Earth's climate can rapidly switch from one state to another, it is imperative that we determine the causes of this variability if we are to accurately predict future climate.

In the original work of Dansgaard (Dansgaard, 1985), the most recent saw-tooth shaped interstadial stadial sequence of climate change since 120,000 years ago, associated with the Bølling-Allerød warming and Younger Dryas stadial (abbreviated here to BA/YD) from ∼14,600 to 11,700 years BP was labeled as D-O event 1 (Figure 1). Since then, however, a growing body of geological evidence attributing the Younger Dryas cooling to a glacial outburst flood and/or a change in glacial meltwater drainage patterns to the ocean (Broecker et al., 1989; Clark et al., 2001; Keigwin et al., 2018) has often led to this episode being treated as a unique event, rather than as one of the D-O stadials (Li and Born, 2019). Evidence that the YD cooling also might have coincided with a meteorite impact capable of 'blocking out' incoming solar radiation has helped bolster this notion (Firestone et al., 2007). In this paper, we use a multivariate outlier method to re-examine the extent to which the BA/YD should be considered 'unique' in the context of the other D-O events. The motivation for this study derives from the remarkable similarities in shape (i.e. deviation from center over its timespan) between the BA/YD and other D-O events within the last 120,000 years, which leads us to question the uniqueness of the BA/YD in the Greenland record. Our approach is particularly novel for traditional paleoclimate research and we argue for the increased implementation of similarly robust statistical methods in future research. Indeed, the application of our statistical method to assessing the 'BA/YD uniqueness' is just one example given that it is exceptionally useful when synthesizing and compare climate records that are both uncertain and complex. It should also be noted that modeling studies suggest that forced and unforced AMOC variations have very similar signatures (Brown and Galbraith, 2016), so the outlier detection technique is not aimed to assess the qualities of D-O events as they result from specific triggers, but rather to provide a framework for situating the BA/YD within a broader context of many other D-O events, each of which may (or may not) have the same underlying trigger.

## 2   Methods

To study abrupt decadal-to-multidecadal changes in climate associated with each of the Dansgaard-Oeschger events, we examined published changes in oxygen isotope ratios ($\delta^{18}O$) and methane ($CH_4$) from the NGRIP Greenland ice cores (Rasmussen et al., 2014; Baumgartner et al., 2014) and $\delta^{18}O$ and carbon dioxide ($CO_2$) changes from the EDML, WAIS, Siple Dome, and TALDICE ice cores recovered from Antarctica (Barbante et al., 2006; Bereiter et al., 2015) that span the last 120,000 years of Earth's climate history. Our EDML record uses the GICC05 age scale, and it was chosen because it has a spatial resolution comparable with the Greenland ice core records. Indeed, the snow accumulation at EDML is two to three times higher than at other deep drilling sites on the East Antarctic plateau, so higher-resolution atmosphere and climate records can be obtained for the last glacial period, making the EDML core especially suitable for studying decadal-to-millennial climate variations in Antarctica. Including EDML $\delta^{18}O$ allows us to observe changes in NGRIP $\delta^{18}O$ as distinct in location but similar in meaning.

This allows us to make conclusions about how the BA/YD may not have been a unique event in Greenland, but perhaps was so in the southern Atlantic. In general, our choice of records is based on those with the highest spatial resolution and tradition in the field of paleoclimatology of using these to study climate variability during both D-O events and the BA/YD. The high temporal resolution of the ice cores during the last glacial period makes them idea for use in our work. Furthermore, we use both $\delta^{18}O$ records (NGRIP and EDML) to provide local approximations of climate, whereas $CH_4$ and $CO_2$ are more indicative of global hydrology and temperature, respectively.

For the purposes of our study, the timing of each Dansgaard-Oeschger event is taken from the ages published in the IN-TIMATE (INTegration of Ice-core, MArine and TErrestrial records) dataset in Table 2 of Rasmussen et al. (2014). We then develop a stratigraphy that emphasizes the large-scale Dansgaard-Oeschger variability as follows: Firstly, as several intersta-dials in the INTIMATE record of Rasmussen et al. (2014) comprise of sub-events labeled by lowercase letters, for our work we consider these to be part of the larger interstadial, and not unique events. For example, while Greenland Interstadial 1 (i.e. the BA interstadial) comprises of sub-events GI-1a through GI-1e, in our analysis this is simply treated as GI-1. A second set of sub-events in the INTIMATE dataset are also denoted by decimals in Rasmussen et al. (2014). For example, Dansgaard-Oeschger event 2 in the INTIMATE dataset is separated into two sub-events, labeled GS 2.1/GI2.1 and GS 2.2/GI 2.2. Due to their generally high amplitude and tendency to span multi-centennial timescales, these sub-events must at least initially be considered as Dansgaard-Oeschger 'candidates', and thus require a more rigorous procedure to be dealt with. Firstly, we consider cases when two sub-events occur in succession and define a duration-based algorithm to determine whether each one should be considered a separate Dansgaard-Oeschger event, both combined into one single event, or omitted from our analysis entirely (Figure 2).

Of the eight Dansgaard-Oeschger events in this period containing two sub-events - namely numbers 2, 5, 15, 16, 17, 19, 21, and 23 - our main analysis leads to the selection of stadial and interstadials found in Table 1. The selection of these events is based on using duration parameter choices: $x = 300$ yrs, $y = 300$ yrs, $z = 200$ yrs, which are at the shorter end of what has previously been accepted as the length of a stadial or interstadial (e.g. Rasmussen et al. (2014)), but our results are not very sensitive to the chosen length. For example, columns 2-3 of Table 1 show that altering these parameters to $(x, y, z) = (90, 100, 140)$ or $(x, y, z) = (90, 100, 90)$ yields results that are 86-93% similar.

Taking D-O event 2 as an example, we observe that GI2.2, GS2.2, and GI2.1 span 120, 200, and 120 years respectively, and thus the algorithm in Figure 2 leads to the combination of GI2.2, GS2.2, and GI2.1 into a single interstadial, since the sub-events are less than the parameter choices $x = 300, y = 300, z = 200$ respectively. In D-O event 5, however, GI5.2, GS5.2, and GI5.1 span 460, 1200, and 240 years, respectively, and thus under the same parameter choices, the interstadial-stadial choice algorithm in Figure 2 dictates that each sub-event should be treated as its own stadial or interstadial. Note that our final results differ minimally based on how sub-cycles are chosen.

Beyond $\sim$104 kyr BP, the $CO_2$ record contains only one data point for about every 500 years. Thus, to ensure the existence of a well-defined and complete record for our chose data, we restrict our analysis of the last glacial cycle to the period of 104-11 kyr BP containing D-O events 1-23. Of the eight containing sub-events, our algorithm discards the second sub-event of four D-O events (i.e., 16.2, 17.2, 21.2, and 23.2), includes two second sub-events as distinct (i.e., 5.2 and 19.2), absorbs GI15.1

into the sub-stadials surrounding it, and absorbs GS2.2 into the sub-interstadials surrounding it (see Table 1 for the algorithm's decisions for other parameter values). This amounts to the consideration of 25 D-O events, four of which are sub-events (i.e., events 5.1, 5.2, 19.1, and 19.2).

To initially examine the uniqueness of the pattern of climate change during Dansgaard-Oeschger event 1 (the BA/YD), we overlaid the NGRIP $\delta^{18}$O record of each D-O event over the BA/YD. To better visualize and compare the shape of each D-O event, we normalized the timescale that covers each D-O event and centered each record at its median. Here the term "normalizing" refers to stretching/shrinking the time over which each D-O event occurred to a consistent number of years, and "centering" to positioning each D-O event in time space by subtracting the median value of each record during each D-O event. We then narrowed down the number of D-O events (including BA/YD) by visually selecting those events that most closely resembled the pattern of NGRIP $\delta^{18}$O during the BA/YD. This process does not apply to the statistical analysis that follows in which all 25 events were included, but as preliminary evidence for the fact that the BA/YD's shape in the context of the Greenland records is not unique in terms of the general shape of many D-O events.

In our second, arguably more important line of analysis, we investigate rigorously the shape (i.e. time evolving variability) of each of our chosen climate records (NGRIP $\delta^{18}$O, EDML $\delta^{18}$O, compiled Antarctic $CO_2$, and NGRIP $CH_4$) during all Dansgaard-Oeschger events (25 including the BA/YD) using a PCOut procedure by calculating (i) the magnitude of change from interstadial to stadial (peak-to-trough analysis), (ii) the rate and direction (slope) of change of each record during each stadial, and (iii) the median value of each record during each stadial. Measurements (ii) and (iii) were considered only for stadial periods due to their known volatility and clear definition in the record. When measurement (ii) was taken on the interstadial data, no significant variation was found. When measurement (iii) was taken on the interstadial data, it mirrored (iii) from the stadial periods, and was thus unnecessary. In our peak-to-trough analysis, we derived a measure of the amplitude of change from the interstadial to the stadial by calculating the difference between the mean of the warmest interstadial points and the mean of the coldest stadial points for each D-O event in the NGRIP $\delta^{18}$O record. To ensure that the peak interstadial warmth and maximum stadial cooling are selected, the mean values are calculated using only the upper and lower 10% of the $\delta^{18}$O values, respectively (Figure 3). We calculate this peak-to-trough measure for the other three records by taking the difference of the mean of its values within the time window of NGRIP $\delta^{18}$O's maximum (minimum) 10% interstadial (stadial) values, acknowledging that some age uncertainty between the records may be present. However, the nature of these age uncertainties is not well known, so we use the aforementioned average of 10% maxima and minima as a robust protection against any age uncertainty. For some records, where no data exists in a given short interstadial (stadial) time window, we take the maximum (minimum) of a 300 year moving gaussian filter (250yr for $CO_2$, in order to give higher weight to each of the sparser points in the dataset), and while not ideal, it is the best approximation that our data limitations can offer.

We estimated the linear slope, and thus overall rate and direction of change, of each record during each of the stadials using ordinary least squares (OLS) regression. In many cases the NGRIP $\delta^{18}$O record behavior during stadial periods is generally flat, so records with highly negative peak-to-trough measurements and stadial slopes close to zero are a good indicator of Dansgaard-Oeschger event behavior. Finally, the median of each record for each stadial in our analysis was calculated. The values of each of these metrics for each record across all 25 chosen D-O events are shown in Tables 2 and 3. It should be noted

that all of the above stated measurements are robust to age uncertainties of at least 100 years. Thus, we can be fairly confident that age and delta age uncertainties will not wildly skew our results.

A robust principal component based outlier detection method, entitled PCOut, based on Filzmoser et al. (2008), was then applied to the results from our three metrics to test if the BA/YD is statistically different from other D-O events. This algorithm is proven to be efficient in high dimensions and especially effective in identifying location outliers, which is ideal for our data. We accept PCOut's slightly higher amount of false positives (i.e., higher size) than other algorithms on the basis that its extremely low level of false negatives (i.e., high power) is more important for this study since the areas in which the Younger Dryas is not unique is of particular interest. PCOut differs from typical principal component analysis schemes in two ways: 1) it robustly transforms the data before extracting principal components, and 2) it computes two measures of variance: one based on location and the other based on scatter. In short, PCOut first shifts an $n \times p$ data array by its variable-wise median and scales it by its variable-wise median absolute deviation (MAD), both of which are more robust (i.e., error resistant) estimators of location and scale (respectively) than sample mean and variance (Filzmoser et al., 2008). In our case, we let $n = 25$ correspond to the number of D-O event observations, and let $p = 12$ variables denote the result of obtaining the three aforementioned metrics on each of the four records (NGRIP $\delta^{18}$O, $CH_4$, and $\delta^{18}$O and $CO_2$ from Antarctica). PCOut then performs a standard principal component analysis (PCA) procedure to the transformed data that retains the first $p^*$ components contributing 99% of the data's variance, and subsequently shifts and rescales the principal components once again by their new median and MAD. For an estimate of location exceptionality, PCOut is programmed to weight each of these resulting components $z_{ij}^*$ by the following robust measure of kurtosis,

$$w_j = \left| \frac{1}{n} \sum_{i=1}^{n} \frac{(z_{ij}^* - \mathrm{med}(z_{1j}^*, \ldots, z_{nj}^*))^4}{\mathrm{MAD}(z_{1j}^*, \ldots, z_{nj}^*)^4} - 3 \right| \text{ for } j = 1, \ldots, p^* \tag{1}$$

and then computes a robust Euclidian distance $RD_i$ for each of the $n$ data points using these weights, where $W = \sum_{j=1}^{p^*} w_j$, the total weight, and the $z_{ij}^*$ are the location shifted and rescaled principal components (visualized in panel 1, Figure 4):

$$RD_i = \sqrt{\sum_{j=1}^{p^*} \left( \frac{z_{ij}^* w_j}{W} \right)^2}. \tag{2}$$

This is followed by a further transformation to acquire the final robust distances $d_i$, where $\chi_{p^*,0.5}^2$ is the 50th percentile of a chi-squared distribution with $p^*$ degrees of freedom:

$$d_i = RD_i \cdot \frac{\sqrt{\chi_{p^*,0.5}^2}}{\mathrm{med}(RD_1, \ldots, RD_n)}. \tag{3}$$

These $d_i$'s represent the degree of separation each of the $n$ data points (corresponding to D-O cycles) experiences from the center of the data, where each $RD_i$ is a robust calculation of the $i$th data vector's distance from its variable-wise median.

Dividing by the median of the $RD_i$'s as in eq. 3 measures how much each $RD_i$ deviates from the median of all such distances. To evaluate these distances as outliers or non-outliers, each data point is assigned a weight $a_i$ based on its distance $d_i$ such that

155 higher distances receive a smaller weight so as to avoid outlier masking (visualized in panel 2, Figure 4):

$$
a_i = \begin{cases} 0 & d_i \geq c \\ \left(1 - \left(\frac{d_i - M}{c - M}\right)^2\right)^2 & M < d_i < c \\ 1 & d_i \leq M \end{cases} .
\tag{4}
$$

For this $a_i$ weight, the parameter $M$ is the $\frac{1}{3}$ quantile of the distances $\{d_i\}$, and

$$
c = \mathrm{med}(d_1, \ldots, d_n) + 2.5 \cdot \mathrm{MAD}(d_1, \ldots, d_n).
\tag{5}
$$

Finally, PCOut defines another metric $b_i$ for each data point that uses the same exact procedure minus the kurtosis weighting

160 step (i.e., unweighted euclidian distance $\sqrt{\sum_{j=1}^{p^*} \left(\frac{z_{ij}^*}{W}\right)^2}$ substitutes for eq. 2). Since the non-kurtosis weighted $d_i$ are proven to follow $\chi_{p^*}^2$ relatively closely, $(M^2, c^2) = (\chi_{p^*, .25}^*, \chi_{p^*, .99}^*)$ in eq. 4's calculation of $b_i$. In the final test (visualized in panel 3, Figure 4), outliers are then defined as data points where

$$
\frac{(a_i + 0.25)(b_i + 0.25)}{(1.25)^2} < 0.25.
\tag{6}
$$

  PCOut achieves much higher precision than a traditional principal component analysis scheme because of the strategic

165 weighting mechanisms aimed to iteratively reduce the degree which outliers mask their own presence (Filzmoser et al., 2008). Further, its use of robust statistical estimators suits our constructed dataset well in that the metrics calculated are subject to high uncertainty. For a graphical representation of PCOut's data transformations spanning eq.'s 2-6, see Figure 4.

  PCOut is not applicable to single variable data because principal component analysis is not a valid procedure for $p = 1$, so outliers in this case are determined using a simpler criterion: if some data point $x_i \notin [Q_1 - \mathrm{MAD}(x_1, \ldots, x_n), Q_3 +$

170 $\mathrm{MAD}(x_1, \ldots, x_n)]$, where $Q_1, Q_3$ are the first and third quantiles of the data, respectively, it is considered an outlier.

## 3 Results

Figure 5 shows that the changes in the NGRIP $\delta^{18}$O record during the BA/YA bear a remarkable similarity to the seven D-O events (namely events 7, 8, 11, 12, 13,16, and 19.2) visually selected. Furthermore, the overlay of mean behavior of all D-O events excluding the BA/YD (i.e. 24 as opposed to just the 7 visually selected) corroborates that the BA/YD period is strikingly

175 similar in NGRIP $\delta^{18}$O shape to the average D-O event. In each case, we observe the classic D-O 'sawtooth' pattern that is characterized by an abrupt warming at the onset of interstadial conditions, followed by a more gradual cooling and return to stadial conditions. These overlays are also completed for the other three records (i.e. $CH_4$ from Greenland, compiled $CO_2$ from

Antarctic, and $\delta^{18}$O from EDML). Beginning with NGRIP CH$_4$, Figure 5 indicates no clear overall pattern to the D-O time series, as the mean line is nearly flat. This confirms a similar lack of uniqueness in BA/YD's CH$_4$ record, which our PCOut analysis will later confirm. The BA/YD appears not to strictly follow the trend of the seven time series lines or mean line in the two Antarctic record overlays (EDML $\delta^{18}$O and CO$_2$), which indicates that further study is required to determine how the BA/YD might constitute an exceptional event from the perspective of these records.

The results from our PCOut analysis allows for additional categorization of the Younger Dryas either as outlier or non-outlier in all variable subsets when equipped with our three metrics applied to four different chemical record for all 25 D-O cycles under consideration. The rationale for observing the Younger Dryas' outlier behavior in particular subsets of the 12-variable system is to understand how record subset (NGRIP $\delta^{18}$O, NGRIP CH$_4$, EDML $\delta^{18}$O, compiled Antarctic CO$_2$), record shape (peak-to-trough, stadial slope), and record location (median), might individually render the the data for this period unique (or not unique). Given PCOut's low level of false negatives compared to other tests of its kind, we take non-outlier results seriously as indicators that Younger Dryas is not statistically exceptional as a D-O event. Relevant results pertaining to the BA/YD are summarized in Table 4, which indicates the subsets of records and metrics for which the BA/YD is an outlier or not - "YES" ("NO") means that the BA/YD is (not) an outlier within that subset. Note we are specifically looking for subsets of records that PCOut identifies as non-outliers, and are not seeking compare the results of such subsets to one another. Thus, the temptation to tally the results should be resisted - this is not a statistically sound way of determine outlier behavior on a large scale.

Beginning with single variable results, we find that all but two cells in the median column (column 4) of Table 4 exhibit outlier behavior. This result is unsurprising given that the BA/YD occurs during a period of overall warming closer to the Holocene and thus higher percentages of all chemical records compared to other D-O events, which all occurred during the coldest stretches of the past 120 kyr. Median measurements for other D-O cycles on the edges of the last glacial period also harbor a proportionally higher level of median measurements due to their temporal proximity to warmer periods before and after the last glacial period. In fact, we find that the this same table of median-only PCOut results for D-O events 2, 20, and 23 harbor $47\%$, $60\%$, and $53\%$ outliers, respectively, which indicates that we can consistently expect subsets of measurements including the median to be greater for D-O events near the beginning and end of the last glacial period. Thus, we attribute the a portion of the BA/YD's outlier behavior in variable subsets including the median to a known temperature increase during the time of its occurrence.

In the single variable stadial slope column (column 3, Table 4), we find particular interest in the fact that all pairs of records including NGRIP $\delta^{18}$O (rows 6-8) do not register as outliers, while all pairs of records not including NGRIP $\delta^{18}$O (rows 9-11) do register as outliers. This exact phenomenon is also reflected in the paired peak-to-peak and stadial slope column (column 1, Table 4), strongly indicating that the presence of NGRIP $\delta^{18}$O within a given variable subset is associated with a lack of outlier behavior of the shape of the BA/YD.

Our main goal is to assess the exceptionality of the BA/YD in the Greenland record. While the single record NGRIP $\delta^{18}$O section (row 12, Table 4) exhibits a mix of measured outliers and non-outliers, it must be taken into account that all variable subsets in this row that cause the BA/YD to become an outlier contain the median measurement, which, as previously stated, contributes significantly to the BA/YD's outlier behavior due to known warming leading up to the Holocene.

The single record NGRIP $CH_4$ row (row 15, Table 4) exhibits no outlier behavior whatsoever across the relevant variable subsets. This record generally follows the shape of its NGRIP $\delta^{18}O$ counterpart, yet often seems to lag or lead $\delta^{18}O$ by centennial timescales (Baumgartner et al., 2014), inevitably causing higher variance in shape metrics chosen in this study. So, a lack of outlier behavior across all NGRIP $CH_4$ subsets primarily indicates that Younger Dryas' lag in $CH_4$ is not unusual. This lack in the NGRIP $CH_4$ single variable median category is also somewhat surprising, and suggests that the magnitude of $CH_4$ amongst all D-O cycles is less closely tied to glacial-interglacial cycles than NGRIP $\delta^{18}O$.

Observing the two NGRIP ($\delta^{18}O$, $CH_4$) records paired across all metric subsets (row 8, Table 4) leads to further interest: namely, no outlier behavior in metrics other than the pure median exists. Since much of the ice-core based knowledge generated on the Younger Dryas relies on these two records, a lack of outlier behavior in shape is a major result, and confirms our analysis of Figure 5. Thus, the fact that the pair of behaviors of $\delta^{18}O$ and $CH_4$ during the BA/YD is not unique compared to the pair of behavior for these records in other D-O events is a stronger conclusion than if we were to restrict this analysis to only one record from Greenland. We observe this in the scatterplots of Figure 6, which plot the value of pairs of metrics for the Greenland shape measurements across all 25 D-O events, and clearly indicate that for each such pair, the BA/YD is within the natural scatter range of all other D-O events. In particular, notice that for the paired peak-to-trough scatterplots (third panel down from first column of Figure 6), the distribution of points roughly forms a ring of which BA/YD is a part, since there is always at least one other point in the plot that is more outlying in either direction. Similarly, in the paired slope scatterplots (fourth panel down from second column of Figure 6), the distribution of points forms a roughly straight line of which the BA/YD is also a part. In fact, these plots provide an excellent example of what an outlier would look like, namely, the point far to the far right of all others in this plot, which turns out to be D-O 23. In sum, if the shape of the BA/YD were an outlier in the Greenland record, these scatter plots would display a clear separation of the BA/YD from all other points in the scatter.

Furthermore, it should be noted that this same PCOut procedure was applied to all other D-O cycles under consideration. No clear trends were found amongst these events as a whole, but two main observations can be made. Firstly, and most importantly, it should be noted that no event exhibited non-outlying behavior isolated within the Greenland shape data as did the BA/YD. This suggests that later research may be successful in proving that the BA/YD's data from Antarctic sources is uniquely shaped as compared to other D-O events. Secondly, we find that most data subsets for D-O events in the middle of our timescale (c. 49-28 ky BP) are overwhelmingly non-outlying, whereas data subsets associated with D-O events on the tails of our timescale are more sporadically outlying and non-outlying. From this we might conclude that the period spanning D-O 3 to D-O 13 generally consisted of regular and "typical" D-O events, whereas D-O events not in this period either have average higher temperature (as previously discussed) or other inconsistencies. This observation does not, however, negate the conclusion that the BA/YD's Greenland data is non-outlying.

Note that we use the stadial/interstadial length parameters $(x, y, z) = (300, 300, 200)$ to choose 25 D-O cycles for this section, but different parameter choices that output 28-30 D-O cycles for analysis (see Table 1) yield results that are 86-93% similar across all D-O cycles. We find this by applying PCOut to all subsets of the 28 and 30 cycle versions of our algorithm output created by implementing the parameters in columns 2 and 3 of Table 1, then comparing the results to our chosen version.

## 4 Discussion

The aim of this study is to precisely and robustly classify the record-based qualities that would render the BA/YD a unique climate event in the context of other abrupt episodes of climate change during the last 120,000 years, known as Dansgaard-Oeschger events. If the BA/YD is to be excluded from the list of D-O events, or assigned its own particular set of triggering mechanisms there must be some statistically sound reason for doing so.

Using four chemical records commonly included in assessments of general D-O behavior - $\delta^{18}O$ and $CH_4$ from NGRIP, Greenland, $\delta^{18}O$ from EDML, Antarctica, and compiled $CO_2$ from multiple Antarctic records - we refrain from performing traditional cross correlation analysis to test for lags, and instead employ a more holistic approach that captures the shape of each D-O cycle in terms of multiple variables. Three measurements to characterize both the location (median) and shape (peak-to-trough difference, stadial slope) of each chemical record for each D-O cycle are taken, and inputted into a robust principal component analysis algorithm (PCOut) to test for outliers.

Our main result is as follows: the observed data for the BA/YD is not a unique compared to that of the other D-O events recorded in the Greenland ice core record, other than the fact that its median $\delta^{18}O$ levels are higher due to its proximity to deglacial warming into the Holocene. The increase in median $\delta^{18}O$ is also not unique to the BA/YD, as D-O events 2, 20, and 23 exhibit a similar phenomenon, which we attribute to their occurrence proximal to long term global climate fluctuations. The non-uniqueness of the BA/YD's shape is clearly indicated by the statistical indistinguishability of the changes in the Greenland ice core record with the other D-O events, especially in terms of its $\delta^{18}O$ variability, for which one-third of other D-O events appear virtually identical (Figure 5). Thus, the BA/YD's data cannot and should not be distinguished from any other D-O cycle in the last glacial period on the basis of Greenland ice core time series shape. In this context, the BA/YD could be understood as a classic example of a D-O event, and deserves further consideration as such when studying the mechanisms that triggered it. The hypothesized meltwater forcing mechanism sometimes invoked for the BA/YD is seldom considered for the other D-O events, and visa versa, the major triggering mechanisms for the D-O events are rarely used to explain the BA/YD. Our results suggest that understanding the causes of the BA/YD would benefit from examining the mechanisms used to explain D-O events, rather than relying on the meltwater hypothesis. Indeed, the role of meltwater forcing in triggering the YD has been questioned a number of times since it was first proposed by W. Broecker and others in 1989. For instance, the YD is widely viewed as a time of glacial re-advance and reduced terrestrial meltwater discharge to the ocean, such that it is likely that freshwater forcing was less during this period (Abdul et al., 2016), making it difficult to explain how the overturning circulation remained weakened for the 1000 year duration of the YD stadial (Renssen et al., 2015). In addition, the termination of the YD, and subsequent rapid warming into the Holocene coincide with a time of increasing meltwater runoff to the North Atlantic (e.g. Fairbanks (1989)) as the Laurentide Ice sheet over North America finally collapsed.

*Data availability.* Measurements for this project can be found in Tables 2 and 3.

*Author contributions.* Author 1 conducted all data analysis, figure creation, and manuscript drafting. Author 2 advised throughout the research process and edited manuscript as necessary.

280 *Competing interests.* The authors declare that they have no conflict of interest.

*Acknowledgements.* This study is made possible by the Summer Student Fellow Program at Woods Hole Oceanographic Institution as well as NSF REU OCE-1852460. Special thanks to Carl Wunsch, Olivier Marchal, and Andy Solow for extra advising.

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

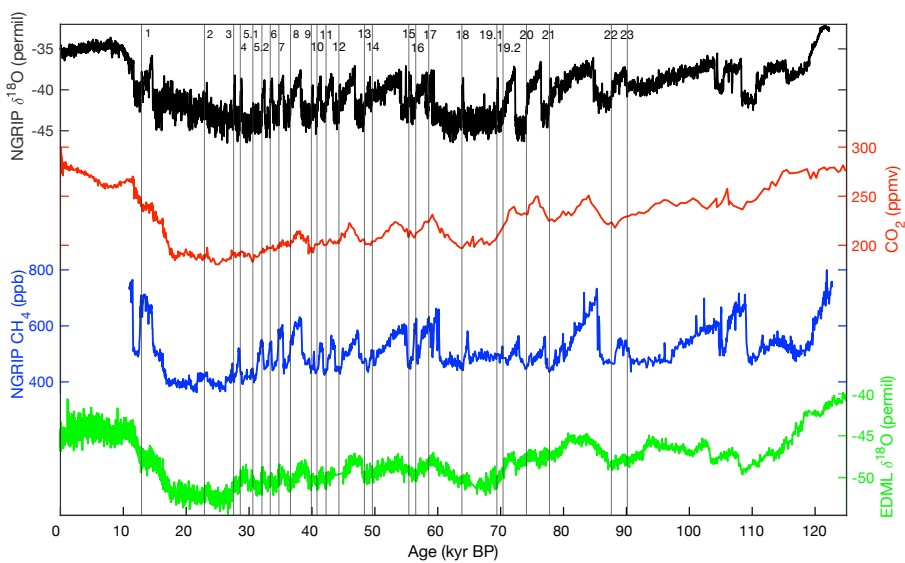

**Figure 1.** Full time series of all four records used. From top down: NGRIP $\delta^{18}$O, compiled $CO_2$ from EDML, WAIS, Siple Dome, and TALDICE, NGRIP $CH_4$, and EDML $\delta^{18}$O. Vertical lines indicate the main 25 interstadial-to-stadial transitions used in this study, labeled by number from Rasmussen et al. (2014).

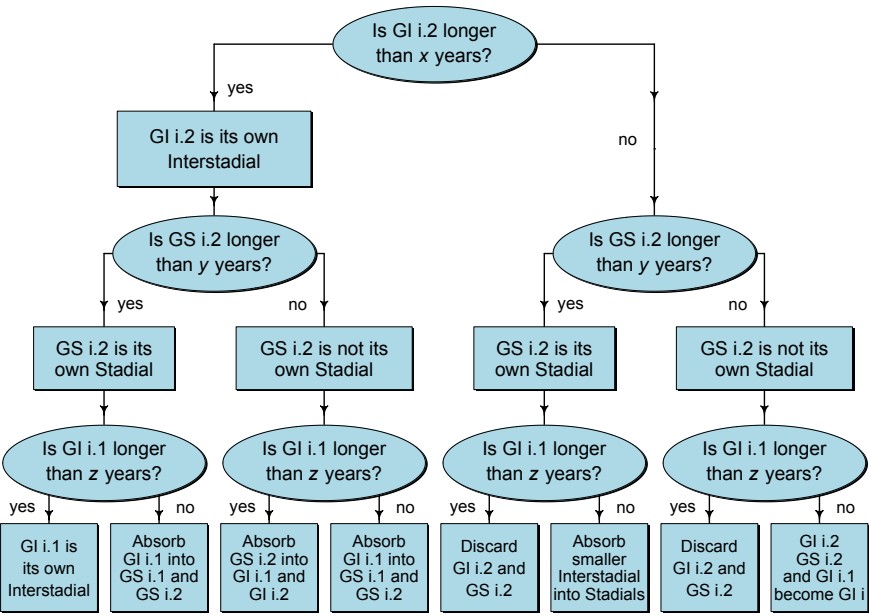

**Figure 2.** Duration-based scheme for including D-O cycles in analysis. Given the variety of climatic shifts present in the high-resolution INTIMATE NGRIP $\delta^{18}$O stratigraphy, it is necessary to form a rigorous criterion for choosing D-O events to analyze. Since the duration of each D-O event in the INTIMATE stratigraphy is directly tied to the confidence that it exhibit the true characteristics of a D-O interstadial (stadial) period, we employ the above decision tree for determining which warm/cold couples split up by the INTIMATE stratigraphy can confidently be considered their own unique D-O events with flexible duration parameters $(x, y, z)$.

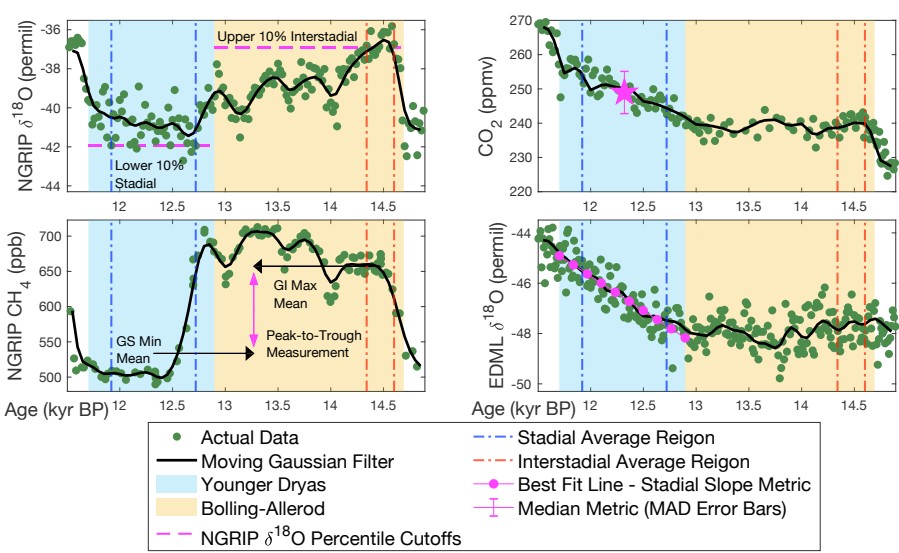

**Figure 3.** Three metrics for capturing the shape and location of D-O cycles across multiple paleoclimate records. In all panels, raw data is presented in green, while interpolated data using a 300yr moving gaussian filter (250yr for $CO_2$, given sparsity of data at points) is presented in black. Blue and orange background shading represents stadial and interstadial conditions, respectively, while pink overlays demonstrate the three measurements taken: in the NGRIP $\delta^{18}O$ panel (upper left), the extreme 10% percentiles of the $\delta^{18}O$ data are determined, and the time window into which all such data falls are extracted (blue (Stadial) and red (Interstadial) vertical average regions). The difference in means of each record's data within these time windows for any given D-O cycle constitutes our peak-to-trough (labeled peak-to-peak) measurement (shown in lower left NGRIP $CH_4$ panel). The OLS linear slope of the stadial data determines the stadial slope for each record in each D-O cycle (shown in the lower right EDML $\delta^{18}O$ panel). The third and final metric measured is the stadial median (shown in the upper right panel). Despite each panel only displaying one metric, all are applied to the four records examined for all D-O events.

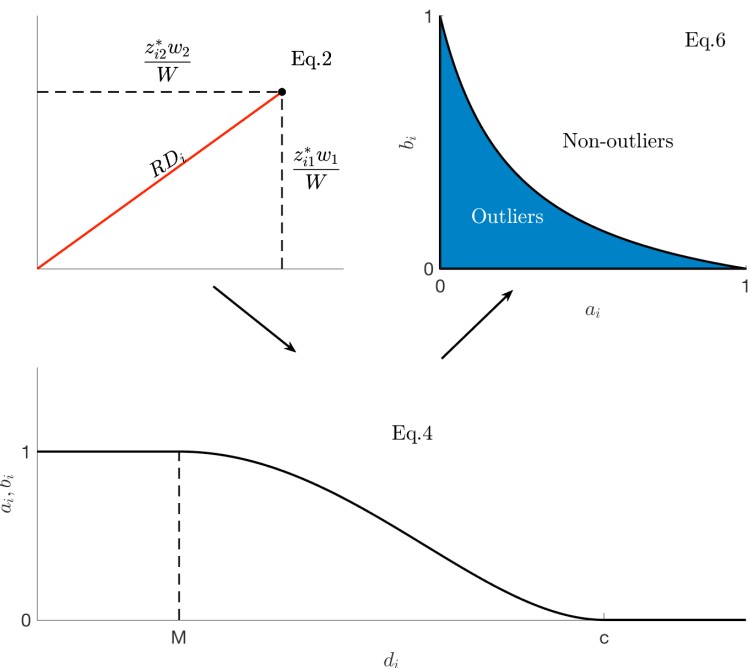

**Figure 4.** PCOut's main outlier decision steps. After completing PCA to produce centered and rescaled components $z_{ij}^*$, eq. 2 calculates the "distance" $RD_i$ of the $i$th component vector from zero with sums of squares (panel 1, top left). After rescaling the $RD_i$'s to create new "distances" $d_i$, we calculate quantities $a_i, b_i$ based on the function in bottom panel 2 (eq. 4), such that large distances $d_i$ translate into smaller values of $a_i, b_i$. Finally, panel 3 (top right) illustrates the region by which PCOut classifies $a_i, b_i$ as indicative of the $i$th datapoint being an outlier or not (eq. 6).

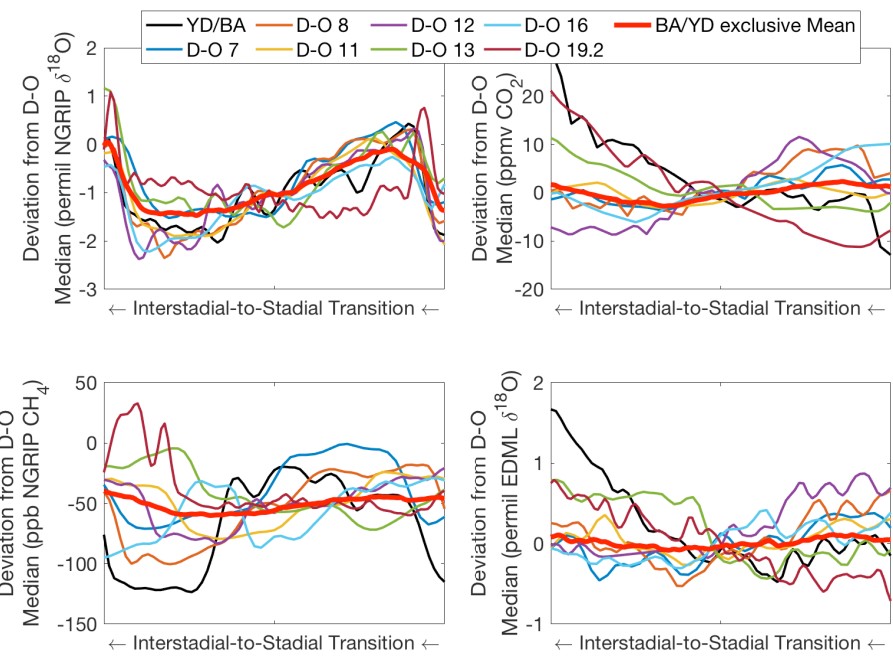

**Figure 5.** Shapes of eight D-O events and grand mean (excluding the BA/YD) with respect to all four records. Representing one-third of D-O events during the last glacial period, these D-O events' respective NGRIP $\delta^{18}$O records bear remarkable similarity to that of the BA/YD, despite assumptions of its uniqueness. Additionally, the BA/YD exclusive mean of D-O events' NGRIP $\delta^{18}$O record confirms that the shape of the BA/YD does not visibly deviate from the classic D-O shape. Further, Antarctic records ($CO_2$ and EDML $\delta^{18}$O, second column) show varying trends that do not appear particularly synchronized to the D-O sawtooth shape. NGRIP $CH_4$ exhibits D-O-like variability, with varying leads and lags. Note: time moves right to left in this figure.

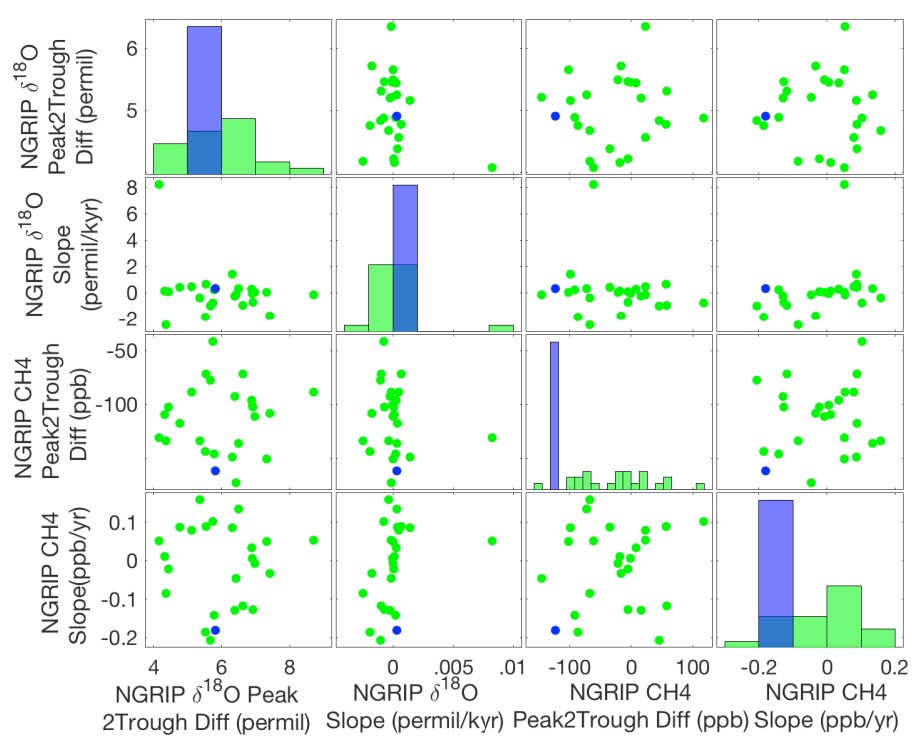

**Figure 6.** Greenland shape scatter grid. Of the 60 multivariate subsets analyzed for outliers, the above represents the essence of our results. Using both peak-to-trough stadial slope and measurements of NGRIP $\delta^{18}$O and CH$_4$ (left), we observe minimal outlier behavior from the BA/YD in each pair of the four variable system, which indicates that the shape of the BA/YD's Greenland records is not unique in of itself. Histograms along the diagonal plot the corresponding single-variable distribution, where the horizontal location of the YD's measurement is in purple on a normalized scale (i.e., the height of the purple bar is 1).

**Table 1.** Three different stadial choice situations for different duration parameter choices in Figure 2. The choice results in the first column $((x, y, z) = (300, 300, 200))$ represent our preferred choices for statistical analysis. Note that basing analysis on the stratigraphic choices represented in the second or third columns yields 86-93% similarity in results.

**Decision Tree Results**

| | | Threshold | |
|---|---|---|---|
| **Cycle #** | $(x, y, z) = (300, 300, 200)$ | $(x, y, z) = (90, 100, 140)$ | $(x, y, z) = (90, 100, 90)$ |
| 2 | Join GI2.1,2.2, and GS2.2 into GI2 | Absorb GI2.1 into GS2.1,2.2 | Treat cycles 2.1 and 2.2 as separate |
| 5 | Treat cycles 5.1 and 5.2 as separate | Treat cycles 5.1 and 5.2 as separate | Treat cycles 5.1 and 5.2 as separate |
| 15 | Absorb GI15.1 into GS15.1,15.2 | Absorb GI15.1 into GS15.1,15.2 | Treat cycles 15.1 and 15.2 as separate |
| 16 | Discard cycle 16.2 | Treat cycles 16.1 and 16.2 as separate | Treat cycles 16.1 and 16.2 as separate |
| 17 | Discard cycle 17.2 | Treat cycles 17.1 and 17.2 as separate | Treat cycles 17.1 and 17.2 as separate |
| 19 | Treat cycles 19.1 and 19.2 as separate | Treat cycles 19.1 and 19.2 as separate | Treat cycles 19.1 and 19.2 as separate |
| 21 | Discard cycle 21.2 | Treat cycles 21.1 and 21.2 as separate | Treat cycles 21.1 and 21.2 as separate |
| 23 | Discard cycle 23.2 | Treat cycles 23.1 and 23.2 as separate | Treat cycles 23.1 and 23.2 as separate |

**Table 2.** Metric measurements for Greenland records. Abbreviations for records are given by $\delta^{18}O$=NGRIP $\delta^{18}O$ and $CH_4$ =NGRIP $CH_4$, and abbreviations for metrics are given by P2T=peak-to-trough, Slp=stadial slope, and Med=median.

| D-O # | $\delta^{18}O$ P2T | $\delta^{18}O$ Slp | $\delta^{18}O$ Med | $CH_4$ P2T | $CH_4$ Slp | $CH_4$ Med |
|---|---|---|---|---|---|---|
| 1 | -5.82 | 0.000341 | -40.73 | -123.59 | -0.181 | 509.80 |
| 2 | -4.33 | 0.000126 | -41.99 | -18.27 | 0.011 | 404.95 |
| 3 | -6.98 | 0.000011 | -43.47 | -22.04 | -0.008 | 400.45 |
| 4 | -6.63 | -0.000958 | -43.95 | 56.95 | -0.118 | 470.40 |
| 5.1 | -4.43 | 0.000090 | -44.28 | -5.33 | -0.022 | 419.00 |
| 5.2 | -6.39 | -0.000228 | -43.69 | 15.37 | -0.129 | 475.40 |
| 6 | -6.93 | -0.000713 | -43.81 | -5.13 | -0.128 | 467.80 |
| 7 | -5.78 | 0.000224 | -42.71 | -91.68 | -0.143 | 467.60 |
| 8 | -6.42 | -0.000144 | -43.02 | -145.68 | -0.046 | 458.80 |
| 9 | -5.12 | 0.000470 | -43.08 | 22.96 | 0.079 | 481.20 |
| 10 | -6.31 | 0.001404 | -42.50 | -98.50 | 0.086 | 457.80 |
| 11 | -5.36 | -0.000372 | -42.29 | -67.87 | 0.158 | 510.50 |
| 12 | -6.5 | 0.000328 | -42.68 | -72.63 | 0.134 | 516.20 |
| 13 | -5.55 | 0.000660 | -42.87 | 56.47 | 0.089 | 532.20 |
| 14 | -4.36 | -0.002479 | -41.52 | -67.58 | -0.086 | 486.55 |
| 15 | -5.75 | -0.000763 | -42.52 | 117.16 | 0.102 | 581.90 |
| 16 | -5.52 | -0.001892 | -42.29 | -87.00 | -0.185 | 574.70 |
| 17 | -5.67 | -0.001038 | -42.04 | 45.40 | -0.207 | 610.10 |
| 18 | -6.89 | 0.000264 | -43.65 | 7.79 | 0.034 | 474.00 |
| 19.1 | -6.9 | -0.000041 | -43.87 | -1.60 | 0.005 | 490.50 |
| 19.2 | -7.43 | -0.001761 | -43.96 | -16.95 | -0.033 | 496.75 |
| 20 | -8.71 | -0.000154 | -44.50 | 23.18 | 0.054 | 519.05 |
| 21 | -7.32 | 0.000036 | -42.59 | -101.45 | 0.051 | 515.15 |
| 22 | -4.76 | 0.000397 | -41.20 | -34.71 | 0.088 | 489.90 |
| 23 | -4.16 | 0.008229 | -39.93 | -61.53 | 0.051 | 512.05 |

**Table 3.** Metric measurements for Antarctic records. Abbreviations for records are given by EDML=EDML $\delta^{18}$O and compiled Antarctic $CO_2$ =$CO_2$, and abbreviations for metrics are given by P2T=peak-to-trough, Slp=stadial slope, and Med=median.

| D-O # | EDML P2T | EDML Slp | EDML Med | $CO_2$ P2T | $CO_2$ Slp | $CO_2$ Med |
|---|---|---|---|---|---|---|
| 1 | 1.12 | 0.002738 | -46.46 | 9.12 | 0.0123 | 248.95 |
| 2 | 0.22 | 0.000534 | -51.40 | 8.10 | 0.0069 | 202.93 |
| 3 | -1.58 | -0.000035 | -52.22 | -6.07 | -0.0006 | 186.14 |
| 4 | -0.85 | -0.00087 | -50.96 | -3.08 | -0.0032 | 189.65 |
| 5.1 | 0.65 | 0.000850 | -50.19 | 5.90 | 0.0044 | 189.83 |
| 5.2 | -1.17 | -0.000367 | -50.96 | -5.08 | -0.0072 | 191.23 |
| 6 | -0.49 | -0.000420 | -50.50 | -0.63 | -0.0050 | 196.14 |
| 7 | -1.19 | -0.000516 | -50.78 | -7.88 | -0.0040 | 196.07 |
| 8 | -1.02 | 0.001384 | -50.20 | -11.34 | 0.0011 | 201.78 |
| 9 | 1.15 | 0.001336 | -49.17 | 11.65 | 0.0072 | 206.09 |
| 10 | -0.42 | 0.000794 | -50.23 | -3.42 | -0.0202 | 197.90 |
| 11 | -0.44 | 0.001256 | -50.08 | -3.21 | 0.0191 | 201.12 |
| 12 | -1.61 | -0.000310 | -49.54 | -15.20 | -0.0003 | 202.59 |
| 13 | 0.92 | 0.001278 | -48.25 | 3.75 | 0.0043 | 206.18 |
| 14 | -1.02 | -0.000076 | -49.60 | -19.76 | -0.0247 | 202.09 |
| 15 | 0.31 | 0.000001 | -48.93 | 1.46 | 0.0085 | 215.38 |
| 16 | -0.83 | 0.000161 | -49.64 | -13.45 | 0.0079 | 208.97 |
| 17 | -0.54 | -0.001232 | -48.54 | -6.58 | -0.0038 | 223.15 |
| 18 | 0.58 | 0.000539 | -49.26 | 8.49 | 0.0060 | 210.57 |
| 19.1 | -0.53 | -0.000012 | -50.66 | -7.24 | -0.0005 | 202.55 |
| 19.2 | -2.28 | -0.001508 | -49.55 | -28.21 | -0.0093 | 213.41 |
| 20 | -0.08 | -0.000665 | -48.16 | -13.03 | 0.0002 | 232.54 |
| 21 | -2.02 | -0.000500 | -47.44 | -8.19 | 0.0141 | 231.68 |
| 22 | 0.23 | 0.000969 | -47.17 | -3.73 | 0.0083 | 226.07 |
| 23 | -1.11 | -0.000438 | -47.98 | -13.36 | 0.0000 | 229.33 |

**Table 4.** PCOut BA/YD Results. "YES" ("NO") indicates that the BA/YD is (not) an outlier in the subset indicated by the row/column combination in which it's located. Rows refer to the record(s) under analysis ($\delta^{18}$O= NGRIP $\delta^{18}$O, compiled Antarctic $CO_2 = CO_2$, EDML= EDML $\delta^{18}$O, and $CH_4$ = NGRIP $CH_4$), and columns refers to the metric(s) applied to those records (P2T=peak-to-trough, Slp= stadial slope, and Med= median). This amounts to an $n \times p$-variate input into PCOut, where $n$ denotes the number of records included and $p$ denotes the metrics applied to all such records.

|    |                                        | 1 | 2 | 3 | 4 |
|----|----------------------------------------|--------|--------|-------|-------|
|    |                                        | (P2T,Slp) | (P2T) | (Slp) | (Med) |
| 1  | ($\delta^{18}$O,$CO_2$,EDML,$CH_4$)    | YES | NO  | YES | NO  |
| 2  | ($\delta^{18}$O,$CO_2$,EDML)           | NO  | NO  | NO  | YES |
| 3  | ($\delta^{18}$O,$CO_2$,$CH_4$)         | NO  | YES | NO  | YES |
| 4  | ($CO_2$,EDML,$CH_4$)                    | YES | YES | YES | YES |
| 5  | ($\delta^{18}$O,EDML,$CH_4$)           | YES | YES | YES | YES |
| 6  | ($\delta^{18}$O,$CO_2$)                | NO  | NO  | NO  | YES |
| 7  | ($\delta^{18}$O,EDML)                  | NO  | YES | NO  | YES |
| 8  | ($\delta^{18}$O,$CH_4$)                | NO  | NO  | NO  | YES |
| 9  | ($CO_2$,EDML)                          | YES | NO  | YES | YES |
| 10 | ($CO_2$,$CH_4$)                        | YES | NO  | YES | YES |
| 11 | (EDML,$CH_4$)                          | YES | YES | YES | YES |
| 12 | ($\delta^{18}$O)                       | NO  | NO  | NO  | YES |
| 13 | ($CO_2$)                               | NO  | NO  | NO  | YES |
| 14 | (EDML)                                 | YES | NO  | YES | YES |
| 15 | ($CH_4$)                               | NO  | NO  | NO  | NO  |