# Peer review of "Assessing the Statistical Uniqueness of the Younger Dryas: A Robust Multivariate Analysis"

_Climate of the Past, 2020_

## Referee Comment (RC1) · Anonymous Referee #1 · 16 Apr 2020

General Comments

In this study, Nye and Condron consider the Bølling-Allerød (BA) and Younger Dryas (YD) in the broader context of abrupt climate change over the last glacial cycle. They apply an outlier detection algorithm to a number of paleoclimate records in order to test whether the BA/YD is statistically unique from DO events of the last glacial. From the results of this outlier detection method, they suggest that the BA/YD is statistically indistinguishable from other DO events (in Greenland ice core records), raising the question of whether its triggering mechanism is unique.

This paper raises important questions regarding our understanding of the mechanisms

printer

of abrupt climate change and applies a novel technique to compare DO events. However, there are several aspects of the paper where more detail/analysis is required. The main components of the paper that I found insufficiently addressed were 1) the use of outlier detection in distinguishing mechanisms of abrupt change, and 2) a more quantitative discussion/demonstration of the (non)uniqueness of the BA/YD, relative to the other 24 considered DO events.

1) Use of outlier detection in distinguishing mechanisms of abrupt change

In this study, Nye and Condron use outlier identification (or non-identification) to 1) argue that the BA/YD should be included in the list of DO events, and 2) suggest that it may not have a unique triggering mechanism (when compared to other DO events). However, the study did not address how outlier detection may be used for this second argument. It is unclear if/how a statistical difference (or more accurately, a similarity) in the selected proxy records would indicate a different (or common) triggering mechanism for these events. As noted by the authors, AMOC variability is often invoked to explain the global signature of DO events and the BA/YD, alike. Modeling studies that compare the global imprint of freshwater forced versus spontaneous AMOC variations (see Brown and Galbraith, 2016, https://doi.org/10.5194/cp-12-1663-2016) suggest that forced and unforced AMOC variations have very similar signatures. This would suggest that similarities between climate proxy records during DO events (and the BA/YD) may not necessarily imply that they were triggered by the same mechanism. Please address the suitability (or limitations) of applying this outlier technique in differentiating between the triggering mechanisms for abrupt climate change.

2) Quantitative discussion/demonstration of non-uniqueness of the BA/YD

The main conclusions of the study are drawn from the results presented in Table 4, which shows the outlier detection results for a given set of climate proxies and metrics. However, from this table it is not obvious that the results support the conclusion that the BA/YD transition is non-unique, given that the BA/YD was identified as an outlier

in most of the tests. It is unclear whether these results are related to the algorithm's relatively high rate of 'false positives' (as the authors mention), or if the BA/YD is actually a statistically unique interval (as defined by the outlier detection algorithm). A fairly simple test of the relative 'non-uniqueness' of the BA/YD would be to perform the same outlier analysis for each of the other 24 events considered in the record.

In Table 4, please include a summary of the results for the other 24 considered DO intervals compared against the 25 DO events. For example, add three columns (and three rows) to the end of the table and include 1) the rate of outlier detections for the BA/YD (for instance, 12/15 or 0.80 for the first column), 2) the average rate of outlier detections for the other 24 events, and 3) the standard deviation of the outlier detections for the other 24 events for each column or row. Including these metrics for how 'unique' the individual DO events are from one another (and the BA/YD) provides a much more direct comparison of the BA/YD to the rest of the DO events. This eliminates the requirement for the reader to have an in-depth knowledge of the nuances of the applied statistical technique to interpret the results for themselves. Without this, it is difficult to assess the 'non-uniqueness' of the BA/YD, and thus the conclusions of the study.

Overall, I think that major revisions are required to provide a convincing argument of how outlier detection may be used to differentiate between mechanisms of abrupt change, and to quantitatively demonstrate the 'non-uniqueness' of the BA/YD. Other aspects of the manuscript that need to be addressed (such as the choice in paleoclimate proxies, and a quantitative assessment of uncertainties) are included in the specific comments. Technical corrections are included in a supplementary document.

Specific Comments

Lines 41-43: The authors do not discuss their choice in which paleoclimate proxies to include in this analysis. Please explain the choice in which proxy records were (and were not) included, and why they are well suited for this analysis. For instance, how are the chosen records better suited in this analysis than other available ice core records for

this interval (such as Greenland/Antarctic aerosol records)? It is also unclear why proxy records from Greenland ice cores are emphasized in this analysis (see comments on lines 178, 189-191). A more thorough discussion of how the chosen records provide insight into the mechanisms/expressions of abrupt climate change would enrich the manuscript.

Line 79: What is the age scale for the EDML d18O? Related to the above questions, why EDML d18O? EDML is often considered unique from other Antarctic ice core records because of its close proximity to the Atlantic basin, but this is not mentioned in the manuscript.

Lines 80-81: How were these three metrics selected? Why were the slopes and medians within the stadial (but not the interstadial) considered?

Lines 85-87: The goal of objectively selecting time windows to compare stadial and interstadial conditions for peak to trough analysis is a worthy one. However, the interval (as defined with water isotopes) may not be appropriate to apply to other variables. For the BA/YD, the selected interval for the stadial (shown in Figure 3, lower left panel) includes the abrupt decrease in CH4, so the amplitude of the peak to trough change appears to be underestimated. This technique assumes that there is no age uncertainty between the selected climate records. Please consider the influence of age (and delta age) uncertainties in selecting stadial and interstadial intervals for peak to trough analysis.

Lines 88-89: Why use a narrower (not wider) filter for CO2 if the data are sparser?

Lines 90-94: Again, error in the alignment of ice core records may influence the median and slope metrics for the selected stadial intervals. Please consider/discuss how age (and delta age) uncertainties may influence these results. It may also be informative to consider how analytical (measurement) uncertainties may affect these metrics (as well as the peak-to-trough metric), and their comparison between DO events.

[Figure]

Lines 95-136: The explanation of the PCOut algorithm is quite detailed, but also important. Please consider shifting some of the details/equations to an appendix.

Lines 146-148: I would caution in generalizing atmospheric $CO_2$ as a Southern Hemisphere proxy, or at least explain the reasoning (also see comments on line 178).

Line 152: It's not totally clear what chemical makeup means here. Does this mean the choice in which proxies are included in the analysis? Please clarify.

Line 178: Why are Greenland proxy records prioritized? See also comments for Lines 189-191. I would also caution in referring to the NGRIP $CH_4$ record as a Greenland proxy. It's true that the record comes from a Greenland ice core, but it is not a proxy for Greenland climate (and is also available from Antarctic ice core records).

Line 183-184: I could be mistaken, but I thought that assessment of leads/lags between $CH_4$ and Greenland temperature came from Baumgartner, 2014, which used d15N-N2 (not d18O) for temperature (so there is no delta age uncertainty).

Lines 189-191: I'm not sure I understand the logic of this argument. Please explain why the NGRIP $CH_4$ and d18O records are particularly well suited to evaluate the (non)uniqueness of the BA/YD in the context of their climatological significance.

Line 201-202: It is unclear how this degree of similarity (86-93%) is quantified. Please specify how the results (with 25 DO cycles versus 28-30) are compared.

Figure 5: It is unclear the direction in which time is moving in this figure.

Table 2: Please check the signs of the metrics. I would expect that the sign for peak-to-trough changes in d18O and $CH_4$ during DO1 (BA/YD) would be the same.

Please also note the supplement to this comment:
https://www.clim-past-discuss.net/cp-2020-43/cp-2020-43-RC1-supplement.pdf

**Supplement:**

**Technical Corrections**

Lines 17-18
Please add a citation for the 8-16°C abrupt warming

Lines 23-24
Please add citations for the three proposed triggering mechanisms that were mentioned

Line 63
Add a space between 460,1200

Line 76
$\delta^{18}$ should be $\delta^{18}O$

Line 103-104
Please include a citation for the statement about the relative robustness of median/MAD as compared to mean/variance.

Line 138
BA/YA should be BA/YD

Line 141
BA/YA should be BA/YD

Line 157
BA/YA should be BA/YD

Line 172
BA/YA should be BA/YD

Line 178
BA/YA should be BA/YD

Line 190
The CH4 and $\delta^{18}O$ records are on the same (NGRIP) ice core. Replace 'cores' with records.

Line 192
BA/YA should be BA/YD

Line 194
BA/YA should be BA/YD

Line 196
BA/YA should be BA/YD

Line 204

BA/YA should be BA/YD

Line 216
BA/YA should be BA/YD

Line 223
BA/YA should be BA/YD

Figure 3
Change 'Reigon' to Region

Figure 3 caption
Metrics should not be capitalized in first line.

Figure 6
$\delta^{18}O$ units should be per mil, not percent

---

## Referee Comment (RC2) · Anonymous Referee #2 · 9 May 2020

Review of cp-2020-43

General comments:

The paper by Nye and Condon applies statistical analysis to metrics derived from ice core data ($\delta$18O, CH4, CO2) spanning a large group of abrupt climate shifts between 120-11 thousand years ago. The procedure PCOut, essentially a form of principal component analysis that is particularly suited for outlier identification, is relatively novel for paleoclimatology studies. The results are interpreted to mean the Bølling-Allerød/ Younger Dryas is not statistically different from 24 other preceding Dansgaard-Oeschger events during the last glacial period in terms of the specific metrics they

derived from the ice core data (specifically, stadial slope and magnitude of peak-to-trough change). The authors conclude that future work should not focus on identifying a unique cause for the YD cold event, suggesting that similar mechanism(s) may have controlled all of the interstadial-to-stadial transitions in the past.

The manuscript is well written, and the analyses and results are very clearly presented. Researchers will be able to repeat this work or apply the techniques to other datasets, thanks to the clear presentation of the methods. I have no issues with the analysis itself, save for a few minor comments listed below. I do, however, think the underlying motivation for the study – the notion that the paleoclimate community considers the cause of the YD event unique from other D-O/ stadial transitions - is overstated throughout the manuscript. Yes, there are studies proposing "one-off" causal mechanisms; for example, there is a large body of literature debating the bolide impact hypothesis. But many researchers think the reason the YD is unique is not because of its cause necessarily, but because it occurred during the last deglaciation, a sort of "failed" transition back to glacial conditions. Or, by the same token, D-O events were "failed" deglaciations that for some reason reverted to glacial conditions. The YD is also unique because similar "reversals" toward glacial conditions did not occur during other terminations of the last 800,000 years. The authors' analysis does not consider these aspects, which to me are more fundamental qualities that make the YD unique, more so than the shapes and slopes of the ice core data themselves. If the authors disagree with this assessment, they should make a stronger case in the introduction for why it should be proven the YD is or is not statistically different from other D-O cycles. More specifically, the authors should specify why their outlier test on only three specific metrics – i.e., stadial slope, median, and peak-to-trough magnitude – is a sufficient test of whether the YD is or is not unique. Is there reason to believe these metrics should look statistically different if the YD was in fact caused by something different than other stadial transitions?

Furthermore, I am not fully convinced that the results support the conclusions. The PCOut results in Table 4 show (at face value, at least) that the YD is statistically signifi-

cantly different from other D-O events. That is, there are 64 instances of yes while only 41 instances of no, and the first column of Table 4 (the all metrics evaluation) points overwhelmingly to outlier status for the BA/YD. The authors' explanation for the median results is that there is significant offset from the rest of the glacial period, but this is precisely why I find the YD is unique and interesting in the first place - not because it looks different from other events in terms of the data, but because it occurred during a deglaciation. That being said, the authors do provide a very clear and thorough discussion of the most interesting aspects of the statistics, which convinced me that the BA/YD is not unique in terms of its expression in NGRIP d18O and CH4. It is useful to point out the statistical likenesses between the BA/YD and other D-O events, as they have done, and I think this paper should be published for this reason, as well as for the reason that the statistical technique is potentially useful for other studies.

I think the manuscript would greatly benefit from revisions such that the paper emphasizes the statistical method rather than the (to me) unsurprising result that the YD is not unique from other stadial events in terms of a few certain patterns resolved in the ice core data. For example, the paper might present the YD/BA is an interesting application of the method, but not an absolute test of YD uniqueness. The description of the method could be bolstered by describing how PCOut might be used for other paleoclimate work, or how it could be adapted to deal with age and measurement uncertainties that are inherent to most paleo datasets. In addition, please see specific comments and technical corrections listed below.

Specific comments:

Line 6-7: Freshwater forcing of circulation due to meltwater from ice sheets or iceberg discharge has been proposed as the cause of stadials during the last glacial period as well, not just for the YD.

Lines 58-59: Can you provide any objective reason for choosing these durations? What makes them conservative?

Line 65: Can you show or say what happens to your final results if you do the extreme scenarios – (1) merge all sub-events into single events, (2) discard all sub-events and only look at the main events, and (3) include all sub-events as their own individual events? This could be in a supplementary section.

Lines 74-77: What exactly do "normalize" and "centered" mean in this context?

Lines 74-77: It is unclear if the "narrowing down" of D-O events applies to the statistical analysis or not. Did you effectively screen the number of events this way? Can you either describe or provide a figure to demonstrate how you narrowed them down, given that this step was done subjectively by eye. Please also state how many events you kept/excluded based on the criteria of visually resembling the BA/YD, assuming you did screen them.

Lines 74-77: Another comment here, and I am assuming that the visual selection of events that you described was used as a screen for the statistical analysis (if not, then disregard the following but please simply reword so it is clear in the text). If so, however, this step would strike me as a major weakness of your analysis. The overall conclusions is that the YD/BA is not statistically distinct from other D-O events in terms of shape, structure, and the other metrics described, but in this selection step you intentionally chose to only look at D-O events that visually resemble the YD/BA in the first place? Please address to what degree the selection criteria influence the final statistical result.

Lines 91-93: Do you have to identify D-O behavior by these criteria for the analysis? I thought you already identified them using the algorithm and the visual resemblance to the BA/YD. If you are just describing the characteristics of D-O behavior, you might change the wording to reflect this so readers are not confused.

Lines 138-140: Again, I don't follow why. Is this just for ease of visualization, or is this related somehow to the statistical analysis? Please state so if that is the case.

Figure 5 – Is the "BA/YD exclusive mean" the mean of all 24 other D-O events, or

[Figure]

just the 7 shown? Please clarify. Additionally, are you sure that the "BA/YD exclusive mean" in the fourth panel (second column, second row) is the mean of EDML d18O? My understanding is this is supposed to be the EDML mean of the 7 events (or 24?), but excluding the BA/YD... It doesn't look like the mean of the other colored lines in that panel, and it looks more sawtooth shaped than the normal phasing of Antarctic temperature with respect to the onset of D-O events. Perhaps there is a mistake.

General technical corrections:

There are numerous places in the text where the term "D-O event" is used somewhat loosely to describe a warming event in the NGRIP ice core record, a cold stadial that follows the warming, or the combination of both warm and cold intervals. Technically the D-O events are just the warming events as they are expressed in the Greenland isotope records.

Specific technical corrections:

Figure 1: The transitions to stadial conditions are marked, but not the transitions to interstadials (i.e. the onset of the D-O events). It is a little confusing when you refer to a specific D-O event. You might consider distinguishing whole interstadial periods versus stadial periods with shading, as in Figure 1 of Rasmussen {Rasmussen, 2014 #751}.

Lines 17-19: Needs a reference.

Line 39: Suggest changing "paleoclimate research" to "future work."

Line 48: Do you mean, "labeled by lowercase letters" in Rasmussen 2014? Please clarify. Line 67: "of a well-defined and complete record for all four of our chosen proxies, as we restrict our analysis of the last glacial cycle to..."

Line 76: "NGRIP d18O," rather than "NGRIP d18"

Line 106: I am being picky here, but two of the four records – NGRIP CH4 and composite CO2 - are not proxies, they are direct measurements.

Line 109: The word "remaining" is confusing to me here? Should it say "resulting," since the components are the result of the principle component analysis?

Equation 1: In reading Filzmoser 2008, I noticed that equation (1) in this manuscript is different from equation (11) in Filzmoser, which has a fourth power in the denominator. However, I also notice the Filzmoser equation (11) is missing a parenthesis in the numerator. Perhaps the fault is in Filzmoser 2008 and not in this manuscript? Please clarify.

Lines 117-120: Consider dividing this long sentence into two sentences for readability.

Equation 4: Would be helpful to define M and c immediately after equation 4, not after describing calculation of bi.

Line 138: I think it should be "BA/YD," not BA/YA. There are other instances of this typo throughout.

Line 145: I disagree with calling the data a CH4 "proxy." A proxy is when you measure one thing, and the data mean something else – like d18O and temperature. In the case of CH4, it's a true measurement of the CH4 concentration in the atmosphere in the past, rather than a proxy for it.

Figure 3: There are some typos in caption and legend.

Figure 4: Consider numbering the panels, or make them look more distinct. I originally thought the top right panel was panel 2. The arrows are not helpful as they are currently displayed because it took me too long to realize they point to the next step in the procedure.

Table 1 caption: "different" rather than "difference." By "point of departure" do you mean "preferred parameter choices used in the statistical analysis?"

[Figure]

[Figure]

---

## Author Response (AR1)

**REVIEWER 1 COMMENTS**
*(Reviewer 2 comments begin on page 9)*

**All text is copied from RC1, and the authors' comments are preceded by "Response:".**

**Updates following reviewer comments are preceded by "Final Revision Update".**

**General Comments**

In this study, Nye and Condron consider the Bølling-Allerød (BA) and Younger Dryas (YD) in the broader context of abrupt climate change over the last glacial cycle. They apply an outlier detection algorithm to a number of paleoclimate records in order to test whether the BA/YD is statistically unique from DO events of the last glacial. From the results of this outlier detection method, they suggest that the BA/YD is statistically indistinguishable from other DO events (in Greenland ice core records), raising the question of whether its triggering mechanism is unique.

This paper raises important questions regarding our understanding of the mechanisms of abrupt climate change and applies a novel technique to compare DO events. However, there are several aspects of the paper where more detail/analysis is required. The main components of the paper that I found insufficiently addressed were 1) the use of outlier detection in distinguishing mechanisms of abrupt change, and 2) a more quantitative discussion/demonstration of the (non)uniqueness of the BA/YD, relative to the other 24 considered DO events.

1) Use of outlier detection in distinguishing mechanisms of abrupt change In this study, Nye and Condron use outlier identification (or non-identification) to 1) argue that the BA/YD should be included in the list of DO events, and 2) suggest that it may not have a unique triggering mechanism (when compared to other DO events). However, the study did not address how outlier detection may be used for this second argument. It is unclear if/how a statistical difference (or more accurately, a similarity) in the selected proxy records would indicate a different (or common) triggering mechanism for these events. As noted by the authors, AMOC variability is often invoked to explain the global signature of DO events and the BA/YD, alike. Modeling studies that compare the global imprint of freshwater forced versus spontaneous AMOC variations (see Brown and Galbraith, 2016, https://doi.org/10.5194/cp-12-1663-2016) suggest that forced and unforced AMOC variations have very similar signatures. This would suggest that similarities between climate proxy records during DO events (and the BA/YD) may not necessarily imply that they were triggered by the same mechanism. Please address the suitability (or limitations) of applying this outlier technique in differentiating between the triggering mechanisms for abrupt climate change.

**Response:** We thank the reviewer for their constructive feedback, and provide some additional details as follows: In our study, we have indeed shown that many proxy-based qualities of the BA/YD are statistically indistinguishable from the other DO events. The fact that the climate response to freshwater forced or spontaneous variations in AMOC is similar is one of key and overarching motivations for our work that a single freshwater forcing mechanism should not be the only triggering mechanism for the BA/YD. Indeed, our main goal with this study is to demonstrate that no assumptions should be made regarding the trigger of the BA/YD, precisely

because it bears strong resemblance to other DO events. In so doing, our revision will make clear that the outlier detection technique is not aimed to assess the qualities of DO events as they result from specific triggers, but rather to provide a general framework for situating the BA/YD within a broader context of many other DO events, each of which may (or may not) have the same underlying trigger. In this sense, our outlier technique is limited to that simple conclusion, and is meant to negate any strong favorability for the BA/YD as triggered by a unique mechanism.

**Final Revision Update:**
- Changed lines 11-12 to emphasize that our results confirm ambiguity of the BA/YD trigger, and highlight a novel statistical method
- Added final sentence of Introduction starting at line 42 to contextualize results
- Changed wording in final paragraph of discussion to emphasize that the proxy *data* is not unique, not necessarily the BA/YD itself (e.g. lines 258 and 264)
* * *
2) Quantitative discussion/demonstration of non-uniqueness of the BA/YD The main conclusions of the study are drawn from the results presented in Table 4, which shows the outlier detection results for a given set of climate proxies and metrics. However, from this table it is not obvious that the results support the conclusion that the BA/YD transition is non-unique, given that the BA/YD was identified as an outlier in most of the tests. It is unclear whether these results are related to the algorithm's relatively high rate of 'false positives' (as the authors mention), or if the BA/YD is actually a statistically unique interval (as defined by the outlier detection algorithm). A fairly simple test of the relative 'non-uniqueness' of the BA/YD would be to perform the same outlier analysis for each of the other 24 events considered in the record.

In Table 4, please include a summary of the results for the other 24 considered DO intervals compared against the 25 DO events. For example, add three columns (and three rows) to the end of the table and include 1) the rate of outlier detections for the BA/YD (for instance, 12/15 or 0.80 for the first column), 2) the average rate of outlier detections for the other 24 events, and 3) the standard deviation of the outlier detections for the other 24 events for each column or row. Including these metrics for how 'unique' the individual DO events are from one another (and the BA/YD) provides a much more direct comparison of the BA/YD to the rest of the DO events. This eliminates the requirement for the reader to have an in-depth knowledge of the nuances of the applied statistical technique to interpret the results for themselves. Without this, it is difficult to assess the 'non-uniqueness' of the BA/YD, and thus the conclusions of the study.

**Response:** Thank you for this comment. While we believe that Table 4 supports our conclusions, we also agree that the way the results are currently presented could be misleading. In our revised manuscript we will make two additional points to clarify the validity of our conclusions from Table 4. Firstly, we will state explicitly that we are specifically looking for subsets of proxies that PCOut identifies as non-outliers, rather than to compare the results of such subsets to one another. This is motivated by sentiments in the paleoclimate community that treat the BA/YD as outlying from other DO events. Secondly, we will make clear that since we are performing tests on all subsets of our data, tallying the total number of subsets registering as outliers is not a statistically sound way of determine outlier behavior on a large scale. Instead, we are asking if

the shape of the BA/YD's Greenland proxies exhibits outlier behavior, for which the answer remains to be no.

We believe that the additional analysis you've proposed is a sound way to add context to our argument, but caution against taking means and standard deviations of the total number of outliers in rows and columns of this data. This is because our goal is not to compare the number of proxy subsets for which the BA/YD is an outlier to other DO events. Rather, our results in Table 4 serve as grounds to support pointed claims about each subset (for example, that the shape of the BA/YD's Greenland proxy data is not an outlier compared to all other DO events). In our revised manuscript, we will certainly perform the PCOut analysis on other DO events as a grounds to discuss the extent to which the general picture of outlier behavior in the BA/YD differs from the other DO events in our study.

We also propose to rework the results section and Table 4 such that the results of PCOut regarding the measurements on proxies of interest (namely, measurements of the shape of Greenland proxies) are highlighted as entirely separate from the other results that indicate the BA/YD as an outlier. This will aid the reader in understanding that the BA/YD may well be an exceptional event in the context of southern hemispheric proxies, but is certainly not when viewed through the Greenland lens.

**Final Revision Update:**
- Modified Table 4 such that only the columns discussed in the results section are present (the other columns were not interpretable or relevant to our conclusions).
- Added lines 191-194 to explicitly state the purpose of the table and irrelevance of summing outlier results.
- Added paragraph from 233-242 discussing conclusion and relevance of performing same analysis on other D-O events.
* * *
Overall, I think that major revisions are required to provide a convincing argument of how outlier detection may be used to differentiate between mechanisms of abrupt change, and to quantitatively demonstrate the 'non-uniqueness' of the BA/YD. Other aspects of the manuscript that need to be addressed (such as the choice in paleoclimate proxies, and a quantitative assessment of uncertainties) are included in the specific comments. Technical corrections are included in a supplementary document.

**Specific Comments**

Lines 41-43: The authors do not discuss their choice in which paleoclimate proxies to include in this analysis. Please explain the choice in which proxy records were (and were not) included, and why they are well suited for this analysis. For instance, how are the chosen records better suited in this analysis than other available ice core records for this interval (such as Greenland/Antarctic aerosol records)? It is also unclear why proxy records from Greenland ice cores are emphasized in this analysis (see comments on lines 178, 189-191). A more thorough discussion of how the

chosen records provide insight into the mechanisms/expressions of abrupt climate change would enrich the manuscript.

**Response:** Our choice of proxies is based on those with the highest spatial resolution and tradition in the field of paleoclimatology of using these to study climate variability during both DO events and the BA/YD. The high temporal resolution of the ice cores during the last glacial period makes them idea for use in our work. Furthermore, we use both d18O records (NGRIP and EDML) to provide local approximations of climate, whereas CH4 and CO2 are more indicative of global climate fluctuations.

**Final Revision Update:**
- Added lines 57-61 to explain choice of proxy.
* * *
Line 79: What is the age scale for the EDML d18O? Related to the above questions, why EDML d18O? EDML is often considered unique from other Antarctic ice core records because of its close proximity to the Atlantic basin, but this is not mentioned in the manuscript.

**Response:** The age scale of the EDML record spans from 150kya to the present. It was chosen because it has a spatial resolution comparable with the Greenland ice core records. Indeed, the snow accumulation at EDML is two to three times higher than at other deep drilling sites on the East Antarctic plateau, so higher-resolution atmosphere and climate records can be obtained for the last glacial period, making the EDML core especially suitable for studying decadal-to-millennial climate variations in Antarctica. Including EDML d18O allows us to observe changes in NGRIP d18O as distinct in location but similar in meaning. This allows us to make conclusions about how the BA/YD may not have been a unique event in Greenland, but perhaps was so in the southern Atlantic. We will certainly make note of EDML's special status as close to the Atlantic basin in our revised manuscript.

**Final Revision Update:**
- Added lines 51-57 to discuss the significance of the using EDML.
* * *
Lines 80-81: How were these three metrics selected? Why were the slopes and medians within the stadial (but not the interstadial) considered?

**Response:** The slope and peak-to-trough metrics are meant to give an idea of the shape of each DO cycle, whereas the median gives us a sense of how cold each stadial became. This latter feature allows us to discuss how larger timescale glacial-to-interglacial changes have the potential to render the BA/YD's average temperature exceptional or not, while the former two (slope and peak-to-trough) allow us to isolate the shape of each DO as an independent feature. We focus solely on the slope and median of the stadial periods because of their known volatility, but will perform an analysis of slope during the interstadials as well in our upcoming revision.

**Final Revision Update:**
- Lines 107-110 were added to explain the selection of measurements taken and the result of analysis of slope on interstadials.
* * *
Lines 85-87: The goal of objectively selecting time windows to compare stadial and interstadial conditions for peak to trough analysis is a worthy one. However, the interval (as defined with water isotopes) may not be appropriate to apply to other variables. For the BA/YD, the selected interval for the stadial (shown in Figure 3, lower left panel) includes the abrupt decrease in CH4, so the amplitude of the peak to trough change appears to be underestimated. This technique assumes that there is no age uncertainty between the selected climate records. Please consider the influence of age (and delta age) uncertainties in selecting stadial and interstadial intervals for peak to trough analysis.

**Response:** Thank you for this comment. In our revised manuscript, we will be sure to explore questions of age uncertainty.

**Final Response Update:**
- Added lines 116-118 to address age uncertainty.
* * *
Lines 88-89: Why use a narrower (not wider) filter for CO2 if the data are sparser?

**Response:** Using a narrower filter for the sparser CO2 data ensures that we are not taking too broad of an average, which, when data is sparse, has the potential to erase important trends in the data. In our revision, we will certainly perform this analysis using a wider filter to quantify this.

**Final Response Update:**
- Lines 119-120: expanded on reasoning for alternate filter.
* * *
Lines 90-94: Again, error in the alignment of ice core records may influence the median and slope metrics for the selected stadial intervals. Please consider/discuss how age (and delta age) uncertainties may influence these results. It may also be informative to consider how analytical (measurement) uncertainties may affect these metrics (as well as the peak-to-trough metric), and their comparison between DO events.

**Response:** We use robust techniques in this paper primarily because of the uncertainties you mention. For example, we take the median of each stadial interval rather than the mean because the mean is far more sensitive to the uncertainties that present themselves when working with proxy data. Further, we take the mean of the top 10% extrema for both interstadials and stadials rather than simply the highest individual value to protect against uncertainty and observationbased variations. In our revised manuscript, we will perform an age uncertainty analysis to see if slight differences in age alter our results significantly.

**Final Response Update:**
- A basic age uncertainty analysis was performed to test whether shifting our data forward or backward in time would skew our measurements. As lines 125-127 suggest, our data is robust to such shifts of at least 100 years (and in some cases, more).
* * *
Lines 95-136: The explanation of the PCOut algorithm is quite detailed, but also important. Please consider shifting some of the details/equations to an appendix.

**Response:** Thank you for your comment. Although equations (1)-(6) regarding PCOut are detailed, we prefer to leave them in the body of the manuscript.

**Final Response Update:**
- No changes made.
* * *
Lines 146-148: I would caution in generalizing atmospheric CO2 as a Southern Hemisphere proxy, or at least explain the reasoning (also see comments on line 178).

**Response:** Noted. In our revised manuscript we will be more specific about the meaning of the CO2 proxy data.

**Final Response Update:**
- Line 182 changed "Southern Hemisphere" to "these proxies".
* * *
Line 152: It's not totally clear what chemical makeup means here. Does this mean the choice in which proxies are included in the analysis? Please clarify.

**Response:** Chemical makeup is incorrect. We mean "record subset", and will amend this in the revised version.

**Final Response Update:**
- Line 186: Change "chemical makeup" to "record subset".
* * *
Line 178: Why are Greenland proxy records prioritized? See also comments for Lines 189-191. I would also caution in referring to the NGRIP CH4 record as a Greenland proxy. It's true that the

record comes from a Greenland ice core, but it is not a proxy for Greenland climate (and is also available from Antarctic ice core records).

**Response:** The Greenland proxy records are prioritized in part for their historical significance in terms of looking at DO events, as well as to address the hypothesis that the climatic signature of the BA/YD and DO events centered predominantly on the North Atlantic due to changes in AMOC. In our revised manuscript, we will be more explicit in discussing the NGRIP CH4 proxy.

**Final Response Update:**
-   See lines 57-61 to explain choice of proxy.
* * *
Line 183-184: I could be mistaken, but I thought that assessment of leads/lags between CH4 and Greenland temperature came from Baumgartner, 2014, which used d15N-N2 (not d18O) for temperature (so there is no delta age uncertainty).

**Response:** Thank you for pointing this out. This citation will be fixed in our revised manuscript.

**Final Response Update:**
- Citation changed from Baumgartner et al. 2012 to Baumgartner et al. 2014 (Line 215).
* * *
Lines 189-191: I'm not sure I understand the logic of this argument. Please explain why the NGRIP CH4 and d18O records are particularly well suited to evaluate the (non)uniqueness of the BA/YD in the context of their climatological significance.

**Response:** The logic of this argument relies on the fact that the pair of these proxies evaluated together are non-outlying behavior across the board. They are both indicators of temperature, yet d18O is generally indicative of local temperature, while CH4 is more global in nature. Thus, the fact that the pair of behaviors of d18O and CH4 during the BA/YD is not unique compared to the pair of behavior for these records in other DO events is a stronger conclusion than if we were to restrict this analysis to only one proxy record from Greenland.

**Final Response Update:**
- Added lines 222-224 to explain reasoning.
* * *
Line 201-202: It is unclear how this degree of similarity (86-93%) is quantified. Please specify how the results (with 25 DO cycles versus 28-30) are compared.

**Response:** For both the 28 and 30 cycle versions of the data, we create a table in the same format as Table 4, and then count which cells display the same result as our Table 4. By this metric,

93% of the cells in the 28 cycle version are the same as the 25 cycle version, and 86% of the cells in the 30 cycle version are the same as the 25 cycle version. We will include text along these lines to our revised manuscript to clarify this analysis

**Final Response Update:**
- Added lines 245-246 to further clarify.
* * *
Figure 5: It is unclear the direction in which time is moving in this figure.

**Response:** Noted. It moves right to left, where more recent times are toward the left. This will also be clarified in our revised manuscript.

**Final Response Update:**
- Clarified in the caption of this figure and the horizontal axes' arrows, which now point left.
* * *
Table 2: Please check the signs of the metrics. I would expect that the sign for peak-to-trough changes in d18O and CH4 during DO1 (BA/YD) would be the same.

**Response:** That's correct, they are the same. We used absolute value for the d18O peak-to-trough measurement because they are all negative, but for the sake of clarity in the revised version we will include the sign.

**Final Response Update:**
-Negative signs added in Table 2 to the d18O P2T column.
* * *
**Technical Corrections:**

**Response:** Thank you for your thoughtful technical corrections. All such corrections will be implemented in our revised manuscript. Citations for lines 17-18 and lines 23-24 are Li and Born (2019), and citation for lines 103-104 will be Filzmoser et al. (2008).

**Final Response Update:**
- All technical corrections implemented.

**REVIEWER 2 COMMENTS**
The paper by Nye and Condon applies statistical analysis to metrics derived from ice core data ($\delta 18O$, CH4, CO2) spanning a large group of abrupt climate shifts between 120-11 thousand years ago. The procedure PCOut, essentially a form of principal component analysis that is particularly suited for outlier identification, is relatively novel for paleoclimatology studies. The results are interpreted to mean the BøllingAllerød/ Younger Dryas is not statistically different from 24 other preceding Dansgaard-Oeschger events during the last glacial period in terms of the specific metrics they C1 CPD Interactive comment Printer-friendly version Discussion paper derived from the ice core data (specifically, stadial slope and magnitude of peak-to-trough change). The authors conclude that future work should not focus on identifying a unique cause for the YD cold event, suggesting that similar mechanism(s) may have controlled all of the interstadial-to-stadial transitions in the past.

The manuscript is well written, and the analyses and results are very clearly presented. Researchers will be able to repeat this work or apply the techniques to other datasets, thanks to the clear presentation of the methods. I have no issues with the analysis itself, save for a few minor comments listed below. I do, however, think the underlying motivation for the study – the notion that the paleoclimate community considers the cause of the YD event unique from other D-O/ stadial transitions - is overstated throughout the manuscript. Yes, there are studies proposing "one-off" causal mechanisms; for example, there is a large body of literature debating the bolide impact hypothesis. But many researchers think the reason the YD is unique is not because of its cause necessarily, but because it occurred during the last deglaciation, a sort of "failed" transition back to glacial conditions. Or, by the same token, D-O events were "failed" deglaciations that for some reason reverted to glacial conditions. The YD is also unique because similar "reversals" toward glacial conditions did not occur during other terminations of the last 800,000 years. The authors' analysis does not consider these aspects, which to me are more fundamental qualities that make the YD unique, more so than the shapes and slopes of the ice core data themselves. If the authors disagree with this assessment, they should make a stronger case in the introduction for why it should be proven the YD is or is not statistically different from other D-O cycles. More specifically, the authors should specify why their outlier test on only three specific metrics – i.e., stadial slope, median, and peak-to-trough magnitude – is a sufficient test of whether the YD is or is not unique. Is there reason to believe these metrics should look

statistically different if the YD was in fact caused by something different than other stadial transitions?

Furthermore, I am not fully convinced that the results support the conclusions. The PCOut results in Table 4 show (at face value, at least) that the YD is statistically significantly different from other D-O events. That is, there are 64 instances of yes while only 41 instances of no, and the first column of Table 4 (the all metrics evaluation) points overwhelmingly to outlier status for the BA/YD. The authors' explanation for the median results is that there is significant offset from the rest of the glacial period, but this is precisely why I find the YD is unique and interesting in the first place - not because it looks different from other events in terms of the data, but because it occurred during a deglaciation. That being said, the authors do provide a very clear and thorough discussion of the most interesting aspects of the statistics, which convinced me that the BA/YD is not unique in terms of its expression in NGRIP d18O and CH4. It is useful to point out the statistical likenesses between the BA/YD and other D-O events, as they have done, and I think this paper should be published for this reason, as well as for the reason that the statistical technique is potentially useful for other studies.

I think the manuscript would greatly benefit from revisions such that the paper emphasizes the statistical method rather than the (to me) unsurprising result that the YD is not unique from other stadial events in terms of a few certain patterns resolved in the ice core data. For example, the paper might present the YD/BA is an interesting application of the method, but not an absolute test of YD uniqueness. The description of the method could be bolstered by describing how PCOut might be used for other paleoclimate work, or how it could be adapted to deal with age and measurement uncertainties that are inherent to most paleo datasets. In addition, please see specific comments and technical corrections listed below.

**Response:** Thank you for your very thoughtful comments. We strongly agree that the result of statistical uniqueness in the NGRIP d18O and CH4 records should be emphasized over the overall uniqueness of the BA/YD with respect to other events. In regards to your first comments, we believe that there is significant treatment of the BA/YD as a unique event in the literature, and seek to broaden the field of considerations for what the BA/YD represents climatologically. In particular, we find it notable that an event so similar in shape to the classic notion of a DO event can occur at a time of deglaciation.

In our revised manuscript, we will emphasize these thoughts in addition to the second meta-purpose of this study, which is to introduce more robust algorithmic outlier detection techniques into the field of paleoclimatology. Indeed, we consider our methods to be incredibly useful when attempting to synthesize and compare climate records that are both uncertain and complex.

**Final Revision Update:**
- Added sentence at lines 40-41.
* * *
**Specific comments:**

Line 6-7: Freshwater forcing of circulation due to meltwater from ice sheets or iceberg discharge has been proposed as the cause of stadials during the last glacial period as well, not just for the YD.

**Response:** Noted, this will be corrected.

**Final Revision Update:**
- De-emphasized uniqueness of trigger hypotheses in line 7.
- Volcanic eruption in line 7 has been deleted.
* * *
Lines 58-59: Can you provide any objective reason for choosing these durations? What makes them conservative?

**Response:** We choose these durations simply because they are at the shorter end of what has previously been accepted as the length of a stadial or interstadial. They are conservative in relation to the Rasmussen et al., 2014 study, which contains many events are that sub-centennial in length. It should also be noted that our results are not very sensitive to the chosen length, and this will be noted in our revision.

**Final Revision Update:**
- Lines 76-80 are added to justify the conservativeness of our choices.
* * *
Line 65: Can you show or say what happens to your final results if you do the extreme scenarios – (1) merge all sub-events into single events, (2) discard all sub-events and only look at the main events, and (3) include all sub-events as their own individual events? This could be in a supplementary section.

**Response:** Yes. We can certainly perform this analysis for inclusion and discussion in the supplementary section of our revised manuscript.

**Final Response Update:**
Upon further reflection, we have decided not to include this additional analysis. Although the results would certainly be interesting, they would not pertain directly to our goal of providing a new method for researchers to implement and replicate. Thank you again for this idea, as it is an interesting question for further research.
* * *
Lines 74-77: What exactly do "normalize" and "centered" mean in this context?

**Response:** These lines refer to the creation of Figure 5. "Centering" these time series at their median means computing and subtracting the median value of each record during each DO event, and "normalizing" means stretching or shrinking the time over which each DO event occurred to a consistent number of years for each. These methods allow us to better visualize and compare the shape of each DO event. In our revision we will make it clear precisely what we are referring to in this context.

**Final Response Update:**
- Added sentence in lines 96-98
* * *
Lines 74-77: It is unclear if the "narrowing down" of D-O events applies to the statistical analysis or not. Did you effectively screen the number of events this way? Can you either describe or provide a figure to demonstrate how you narrowed them down, given that this step was done subjectively by eye. Please also state how many events you kept/excluded based on the criteria of visually resembling the BA/YD, assuming you did screen them.

**Response:** This process does not apply to the statistical analysis that follows it, and we will make this clear in our revision. But, rather, this serves as preliminary evidence for the fact that the BA/YD's shape in the context of the Greenland records is not unique in terms of the general shape of many DO events. Visually selecting 7 out of 25 DO events that appear indistinguishable in shape from the BA/YD evidences a similarity and a motivation for the following statistical study. We are aware that performing our subsequent analysis on only 8 DO events is not a rigorous test, so use all 25 DO events.

**Final Response Update:**
   - Added sentence in lines 100-102 to clarify.
   - Added clause at the beginning of line 103.
* * *
Lines 74-77: Another comment here, and I am assuming that the visual selection of events that you described was used as a screen for the statistical analysis (if not, then disregard the following but please simply reword so it is clear in the text). If so, however, this step would strike me as a major weakness of your analysis. The overall conclusions is that the YD/BA is not statistically distinct from other D-O events in terms of shape, structure, and the other metrics described, but in this selection step you intentionally chose to only look at D-O events that visually resemble the YD/BA in the first place? Please address to what degree the selection criteria influence the final statistical result.

**Response:** N/A, it is not a screen. Our revised manuscript will make sure this section of our methods is much clearer.

**Final Revision Update:**
- See above revision.
* * *
Lines 91-93: Do you have to identify D-O behavior by these criteria for the analysis? I thought you already identified them using the algorithm and the visual resemblance to the BA/YD. If you are just describing the characteristics of D-O behavior, you might change the wording to reflect this so readers are not confused.

**Response:** Our methods follow two strains of logic: the first concerns the visual selection of 7 DO events to provide evidence for the BA/YD's visual similarity to nearly one third of other DO events studied. Our second, more important line of analysis is that which takes the peak-to-trough, slope, and median measurement in order to implement the PCOut procedure. Both of these analyses are preceded by the algorithmic selection of DO events found in Figure 2. The distinction between different parts of our methods will be made much more clear in the revised manuscript.

**Final Revision Update:**
- See above revision.
- This is further clarified in the altered wording in lines 104-105.
* * *
Lines 138-140: Again, I don't follow why. Is this just for ease of visualization, or is this related somehow to the statistical analysis? Please state so if that is the case.

**Response:** Yes, correct. We will make it explicitly clear in our revision that this is just for ease of visualization.

**Final Revision Update:**
- Lines 171-180 edited to make the separateness of this visual analysis clear.
* * *
Figure 5 – Is the "BA/YD exclusive mean" the mean of all 24 other D-O events, or just the 7 shown? Please clarify. Additionally, are you sure that the "BA/YD exclusive mean" in the fourth panel (second column, second row) is the mean of EDML d18O? My understanding is this is supposed to be the EDML mean of the 7 events (or 24?), but excluding the BA/YD. . . It doesn't look like the mean of the other colored lines in that panel, and it looks more sawtooth shaped than the normal phasing of Antarctic temperature with respect to the onset of D-O events. Perhaps there is a mistake.

**Response:** The "BA/YD exclusive mean" is the mean of just the 7 D-O events shown, and our revised manuscript will explore the possibility of changing this mean to reflect the average of all

24 other DO events. The mean of the EDML is incorrect, thank you for spotting that. It will be corrected in our revision.

**Final Revision Update:**
- Figure 5 is corrected as stated above. The average is now the average of all 24 D-O events, excluding the BA/YD.
- This figure with 24 D-O events is far too cluttered to include, so we stick to our current analysis.
* * *
**General technical corrections:**

There are numerous places in the text where the term "D-O event" is used somewhat loosely to describe a warming event in the NGRIP ice core record, a cold stadial that follows the warming, or the combination of both warm and cold intervals. Technically the D-O events are just the warming events as they are expressed in the Greenland isotope records.

**Response:** Noted, these will be corrected such that D-O events only refer to the warmings.

**Final Revision Update:**
- Multiple instances of this error have been corrected.
* * *
**Specific technical corrections:**

Figure 1: The transitions to stadial conditions are marked, but not the transitions to interstadials (i.e. the onset of the D-O events). It is a little confusing when you refer to a specific D-O event. You might consider distinguishing whole interstadial periods versus stadial periods with shading, as in Figure 1 of Rasmussen {Rasmussen, 2014 #751}.

**Response:** Thank you for this note. For our revised manuscript we will look at different ways of plotting the data so that interstadial periods and stadial periods are easily distinguished.

**Final Response Update:**
- For ease of viewing, our Figure 1 is left unchanged. After experimenting with shading, we find that it is not viable for a small graph like Figure 1, and prefer to mark the transitions as before.
* * *
Lines 17-19: Needs a reference.

**Response:** Indeed, thank you. In our revision, we will reference Li and Born (2019) for this.

**Final Revision Update:**

- Li and Born (2019) is referenced for lines 17-19.
* * *
Line 39: Suggest changing "paleoclimate research" to "future work."

**Response:** Yes, this phrase will be altered in our revised manuscript.

**Final Response Update:**
- This has been rephrased in line 40.
* * *
Line 48: Do you mean, "labeled by lowercase letters" in Rasmussen 2014? Please clarify. Line 67: "of a well-defined and complete record for all four of our chosen proxies, as we restrict our analysis of the last glacial cycle to. . ."

**Response:** Yes, this is what we mean. We'll make clear that this labeling comes from the Rasmussen 2014 paper.

**Final Response Update:**
- This has been made clear in line 65.
* * *
Line 76: "NGRIP d18O," rather than "NGRIP d18"

**Response:** Thank you, this will be changed.

**Final Response Update:**
- This has edited in line 95.
* * *
Line 106: I am being picky here, but two of the four records – NGRIP CH4 and composite CO2 - are not proxies, they are direct measurements.

**Response:** Correct. The revised manuscript will reflect the correct language for these data.

**Final Response Update:**
- Line 139 reflects this change.
* * *
Line 109: The word "remaining" is confusing to me here? Should it say "resulting," since the components are the result of the principle component analysis?

**Response:** Indeed, "resulting" is more intuitive. This change will be made.

**Final Response Update:**
- Line 142 reflects this change.
* * *
Equation 1: In reading Filzmoser 2008, I noticed that equation (1) in this manuscript is different from equation (11) in Filzmoser, which has a fourth power in the denominator. However, I also notice the Filzmoser equation (11) is missing a parenthesis in the numerator. Perhaps the fault is in Filzmoser 2008 and not in this manuscript? Please clarify.

**Response:** The missing parenthesis in the numerator of Filmozer equation (11) should indeed be present. We have, however, forgotten to include the fourth power in our equation (1) denominator. Thank you for noticing, this will be corrected.

**Final Response Update:**
- This correction has been made in line 144.
* * *
Lines 117-120: Consider dividing this long sentence into two sentences for readability.

**Response:** Certainly, thank you.

**Final Response Update:**
- Line 153 reflects this change.
* * *
Equation 4: Would be helpful to define M and c immediately after equation 4, not after describing calculation of bi.

**Response:** Thank you, this will be changed.

**Final Response Update:**
- This edit can be found in lines 159-162.
* * *
Line 138: I think it should be "BA/YD," not BA/YA. There are other instances of this typo throughout.

**Response:** This typo will be corrected everywhere it occurs, thank you.

**Final Response Update:**
- These typos have all been corrected.
* * *
Line 145: I disagree with calling the data a CH4 "proxy." A proxy is when you measure one thing, and the data mean something else – like d18O and temperature. In the case of CH4, it's a true measurement of the CH4 concentration in the atmosphere in the past, rather than a proxy for it.

**Response:** Indeed, thank you for this clarification. Our revised language will better reflect this fact.

**Final Response Update:**
- These errors have all been corrected.
* * *
Figure 3: There are some typos in caption and legend.

**Response:** Noted, thank you. These will be corrected in our revision.

**Final Revision Update:**
- Caption and legend have been corrected for typos.
* * *
Figure 4: Consider numbering the panels, or make them look more distinct. I originally thought the top right panel was panel 2. The arrows are not helpful as they are currently displayed because it took me too long to realize they point to the next step in the procedure. Table 1 caption: "different" rather than "difference." By "point of departure" do you mean "preferred parameter choices used in the statistical analysis?"

**Response:** For our revised manuscript we will produce a clearer version of Figure 4; in particular we will emphasize the fact that each panel in Figure 4 is labeled by the equation it corresponds to, and change the "point of departure" phase such that it clearly describes our preferences.

**Final Response Update:**
-    The recommended edits have been made for Table 1's caption

- The language of figure 4's caption has been changed to make clear that the figure contains 3 panels, each of which corresponds to an equation in our methods section.

**Assessing the Statistical Uniqueness of the Younger Dryas: A Robust Multivariate Analysis**

Henry Nye[1] and Alan Condron[2]

[1]Department of Mathematics and Statistics, Haverford College, PA 19041
[2]Geology and Geophysics, Woods Hole Oceanographic Institution, MA 02543

**Correspondence:** Henry Nye (hnye@haverford.edu), Alan Condron (acondron@whoi.edu)

**Abstract.** During the last glacial period (c. 120-11 kyr BP), dramatic temperature swings, known as Dansgaard-Oeschger (D-O) events, are clearly manifest in high resolution oxygen isotope records from the Greenland ice sheet. Although variability in the Atlantic Meridional Overturning Circulation (AMOC) is often invoked, a unified explanation for what caused these 'sawtooth shaped' climate patterns has yet to be accepted. Of particular interest is the most recent D-O shaped climate pattern that occurred from ∼14,600 to 11,500 years ago - the Bølling/Allerød (BA) warm interstadial and the subsequent Younger Dryas (YD) cold stadial. Unlike earlier D-O stadials, the YD is frequently considered a unique event, potentially resulting from a rerouting and/or flood of glacial meltwater into the North Atlantic or a meteorite impact, or a volcanic eruption. Yet, these mechanisms are seldom less frequently considered as the cause of the earlier stadials. Using a robust multivariate outlier detection scheme - a novel approach for traditional paleoclimate research - we show that the pattern of climate change during the BA/YD is not statistically different from the other D-O events in the Greenland record, and that it should not necessarily be considered unique when investigating the drivers of abrupt climate change. Our results thus confirm the ambiguity of the BA/YD's trigger and present a novel statistical framework for paleoclimatic data analysis. Our results thus raise important questions about the ability of glacial meltwater input and other 'one off' events to trigger abrupt, centennial-to-millennial length, changes in climate

*Copyright statement.* TEXT

[revised manuscript text omitted]

---

## Author Response (AR2)

**Ed Brook 2/10/2021 Comments Response**

**Reply to Comments:** Thank you for your thoughtful comments. We, the authors have elected to implement all the changes you recommend, aside from one. The following list details those changes, and line numbers refer to the marked-up version "NyeCondon_CP_2ndMarkup.pdf".
* * *
**Line 11:** Edited final sentence of abstract to eliminate the statement that our results "confirm the ambiguity of the BA/YD's trigger".

**Line 19:** Added Rasmussen et al., 2014 citation.

**Line 32:** Changed "often" to "sometimes".

**Line 37:** Relegated definition of "shape" to a more relevant section.

**Lines 39-42:** Made motivation of novel statistical methods more concise.

**Lines 52-53:** Clarified that records use the GICC05 age scale, and that record length was a key factor in our decision to use the chosen records.

**Lines 61-64:** Clarified the meaning of CH4 and CO2 records.

**Lines 67-69:** Clarified our interpretation of small sub-events.

**Lines 76-77:** Introduced the meaning of Figure 2 in its own sentence.

**Line 92:** Eliminated and clarified the confusing phrase "containing D-O events".

**Lines 100-105:** Simplified the description of "normalizing" and "centering" and clarified the associated analysis' relationship to our main analysis.

**Lines 109-110:** Summarized PCOut upon its first introduction, rather than later in the paper.

**Lines 113-116:** Clarified the omission of measurements (ii) and (iii) on interstadial data; corrected small syntax errors.

**Line 123:** Typo correction

**Line 125:** Started new sentence at "While not ideal…"

**Lines 127-129:** Deleted extraneous sentence.

**Line 133:** Moved summarization of PCOut to lines 109-110.

**Line 136:** Changed "our data" to "the data considered here".

**Line 139:** Deleted extraneous word, "robustly".

**Line 140:** Changed "location" to "center".

**Line 164:** Clarified that the above procedure applies to $a_i$.

**Line 166:** Clarified that eq. 4 is used to calculated $b_i$ later, but it does not apply to $b_i$ as written.

**Lines 169-171:** Provides basic intuition for weights $a_i$ and $b_i$ and clarifies the use of 0.25 and 1.25 in eq. 6.

**Line 174:** Changed "constructed dataset" to "study".

**Lines 178-179:** Replaced mathematical notation with more readable explanation of single variable procedure.

**Line 185:** Changed "compiled" to "composite".

**Line 186:** Typo correction

**Lines 228-229:** Deleted incorrect statement regarding the volume of records used to study the YD.

**Lines 244-245:** Deleted extraneous sentence.

**Lines 245 and 247:** Changed "our" to "the observed".

**Lines 258-259:** Deleted extraneous sentence.

**Lines 261-262:** Deleted statement about cross-correlation lag analysis.

**Line 268:** Changed "increase in" to "higher".

**Lines 275-277:** Deleted extraneous sentence.

**Lines 285-286:** Clarified that metric measurements are found in our tables, and that the data used can be found in the given sources.

**Line 287:** Clarified author names.

**Figure 1:** Added references in caption. Dr. Brook made an additional suggestion to indicate stadial regions, which we elect to respectfully decline. We remain convinced that further additions to this figure would overcrowd it, and that indicating stadials would not increase the readability of our method from this figure. Instead, we note that the vertical lines show the interstadial-stadial transitions, and thus mark the beginning of each stadial.

**Figure 2:** Initial sentence of caption is now descriptive, whereas the final sentence is now more explanatory.

**Figure 3:** Corrected typos, clarified pink arrows and symbols, and clarified the visual representations of metrics in relation to the actual analysis performed.

**Figure 4:** This figure has been altered to better represent its depiction of the PCOut procedure. Equation numbers are now closer to the graphs representing them, and arrows are longer and more indicative of the fact that equations are being sequentially transitioned between. Changes in the caption also align with these clarifications in order to better aid the reader in understanding this figure: The first sentence is more descriptive, and panels are more clearly referred to.

**Tables 2&3:** These tables' referral to stadials rather than D-O events is now clearer, both in the caption and the table itself.